# Untangling the network effects of productivity and prominence among scientists

**Weihua Li** [1,2,3,4] ✉, **Sam Zhang** [5], **Zhiming Zheng**[1,2,3,4], **Skyler J. Cranmer**[6] & **Aaron Clauset** [7,8,9] ✉

While inequalities in science are common, most efforts to understand them treat scientists as isolated individuals, ignoring the network effects of collaboration. Here, we develop models that untangle the network effects of productivity defined as paper counts, and prominence referring to high-impact publications, of individual scientists from their collaboration networks. We find that gendered differences in the productivity and prominence of mid-career researchers can be largely explained by differences in their coauthorship networks. Hence, collaboration networks act as a form of social capital, and we find evidence of their transferability from senior to junior collaborators, with benefits that decay as researchers age. Collaboration network effects can also explain a large proportion of the productivity and prominence advantages held by researchers at prestigious institutions. These results highlight a substantial role of social networks in driving inequalities in science, and suggest that collaboration networks represent an important form of unequally distributed social capital that shapes who makes what scientific discoveries.

Scientific discoveries are an emergent phenomenon of the collective actions of individual scientists and the scientific communities they construct. The composition of these communities can be highly heterogeneous, and often exhibit pervasive inequalities. These inequalities can be social in terms of who makes up the scientific workforce[1,2] and what resources they receive for their research[3–5], or epistemic in terms of which ideas spread further and receive more attention[6,7]. Understanding the origins of these inequalities and their effects on the pace and direction of scientific discovery would better inform efforts to support innovation, broaden participation in science, and accelerate new discoveries[8].

The pervasiveness of inequalities in science, in representation, prestige, attention, resources, etc., likely reflects the combined and heterogeneous effects of many processes, including competition, cumulative advantage, systemic bias, pipeline effects, and discrimination. For instance, in the academic job market, faculty hiring committees tend to hire the graduates of prestigious doctoral programs[1,9], which may allow scientists at a small group of elite institutions to effectively set the research agenda of the entire field. Scientists at elite institutions also receive disproportionately more funding than those at less prestigious institutions, which may enable greater scientific activity, larger doctoral training programs, and institutionalized hierarchy[10]. And, an elite affiliation provides a measurable advantage in peer review, which may play a role in the research of elite scientists being far more likely to appear in high-impact publication venues, compared to that of early-career or non-elite scientists[11].

Biases related to gender, race, ethnicity, geography, language, and prestige are known to drive differences in scientific output and impact.

[1]LMIB, NLSDE, BDBC, and Institute of Artificial Intelligence, Beihang University, 100191 Beijing, China. [2]Peng Cheng Laboratory, 518055 Shenzhen, China. [3]Zhong Guan Cun Laboratory, 100080 Beijing, China. [4]Beijing Academy of Blockchain and Edge Computing, 100080 Beijing, China. [5]Department of Applied Mathematics, University of Colorado Boulder, Boulder, CO 80309, USA. [6]Department of Political Science, The Ohio State University, Columbus, OH 43210, USA. [7]Department of Computer Science, University of Colorado, Boulder, CO 80309, USA. [8]BioFrontiers Institute, University of Colorado, Boulder, CO 80303, USA. [9]Santa Fe Institute, Santa Fe, NM 87501, USA. ✉e-mail: weihuali@buaa.edu.cn; aaron.clauset@colorado.edu

A number of recent studies have shown that inequality in social networks and collaborations may relate to gender disparity and affect career outcomes for women[12–18], particularly in science, technology, engineering, and mathematics (STEM) fields[19–22]. Moreover, women tend to receive less funding[23], publish fewer papers, are more isolated in collaborations, and are often overlooked in favor of male collaborators[24]. As a result, it remains unclear the degree to which differences in individual scientific activity reflect genuine differences in scientific merit or biases caused by various non-meritocratic processes.

At its base, science is composed of networks of social interactions[25–28]. These interactions mediate most scientific activities, including scientific training, hiring, collaboration, teaching, citation, peer review, and debate. Hence, a scientist's social relationships with other scientists may represent a form of persistent social capital that can be accumulated, used, and possibly transferred among scientists[29–31]. For instance, some evidence indicates that a single extremely strong connection to another scientist is sufficient to increase the productivity and career sustainability of individual researchers[32]. Via collaboration, networks correlate with the unequal provision of "scientific and technological human capital" across researchers[20], shape the academic career of researchers[33], and can conceal underlying inequalities in formal evaluations like tenure[34]. Even common but unadjusted measures of scientific productivity and impact, such as the number of papers a scientist publishes or the number of citations a paper receives, depend on networks, because discoveries are always situated within a broader, evolving conversation among scientists[35–38].

In the sociology of science, there are both many measures of scholarly output and a wide variety of normalization schemes intended to help distill a collaborative publication record into individual-level contributions[8]. For instance, authorship may be fractionalized by the number of coauthors on a given paper[39,40], or a paper's citation count may be normalized by the impact factor of the venue in which it appeared[41]. Each measure sheds its own light on social and epistemic inequalities in science, and each normalization scheme comes with assumptions, with potentially uncertain external validity[42]. In this study, our analysis follows the long tradition in the sociology of science[43,44] of using simple measures of scholarly productivity and prominence, which count the number of papers published by an individual and the number of publications in high-impact venues. This approach presents both advantages and limitations, which we discuss below, but is central to our analysis of network effects.

By mediating scientific attention, evaluation, and collaboration, social networks play a fundamental role both in shaping what scientific discoveries are made and what impact they have, and in shaping pervasive social and epistemic inequalities in science. Untangling the effects of these interactions would shed substantial light on the mechanisms that underlie scientific discovery, and may offer new solutions for making the scientific community more inclusive and innovative. For instance, is it more important for an early career scientist to have a prominent mentor or to train in an elite program? How does who a scientist knows shape what questions they study or what discoveries they make? How much are gender differences in productivity and prominence caused by gendered differences in collaboration networks?[12] And, how much of a scientist's productivity and prominence is explained by that of their collaborators?[45] These questions cannot be clearly answered without considering the effects of social networks in science.

Here, we untangle the network effects of collaborations on the productivity and prominence of individual scientists by developing two network models. Applied to large-scale scientific publication and collaboration data, these models allow us to quantify the network's effect on driving certain widespread and persistent inequalities across individual researchers. Using these models, we investigate the degree to which gendered collaboration patterns explain gendered

differences in productivity—measured by the number of first- or last-authored publications—and prominence—measured by the number of high-impact publications that received the upper 8th percentile of citations as measured 2 years after publication for a given year and field, how network effects vary with institutional prestige, and the degree to which collaboration networks operate as a kind of moderately transferable form of social capital, by which successful senior scientists improve the long-term trajectory of their junior collaborators. Although the selected metrics of productivity and prominence are broadly mentioned and discussed in the scientific community, they should be used with considerations as they do not necessarily imply scientific utility[46,47].

## Results
We begin by extracting pairs of coauthors defined across 20.0 million research articles in the Microsoft Academic Graph (MAG) database since 1950[48,49], across six STEM fields: biology, chemistry, computer science, mathematics, medicine, and physics. To better isolate the most important network connections, we focus on the coauthorship links defined by the first and last authors of each paper. Subsetting to only the first-last author pairs connections eliminates the network effects on productivity and prominence caused by variations in the number of coauthors per paper, middle-author contributions of all types, trends over time and across fields in team sizes, and other related confounds. This selection preserves and focuses our analysis on the most important collaboration links according to common coauthorship norms in STEM fields, e.g., traditional mentor-mentee relationships, where the junior scholar is typically the first author and their senior colleague is the last author.

The nature of coauthorship in scientific publications tends to confound direct measures of the productivity and prominence of individual scientists. Highly productive scientists tend to have many collaborators, often including each other, and the productivity of these individuals tends to lift the productivities of others by virtue of those collaborations. In the same way, highly cited scientists tend to increase the prominence of their collaborators, and often, the same collaborators are both highly productive and highly cited. Bibliometric normalization schemes, such as fractional authorship, can be viewed as paper-level adjustments for these network effects of collaboration.

However, untangling the network effects of collaborations over a scientific career to estimate each individual's contributions within the interdependent context of coauthorship networks requires a generative network model. Here, we introduce two such models that can control for these collaboration network effects and allow us to quantify the latent productivity and prominence of individual researchers, and their relationship with social and epistemic inequalities in scientific careers.

We model the production of publications by a pair of coauthors as a stochastic outcome of their joint efforts, governed by a linear combination of their individual latent productivity parameters (Fig. 1a). Mathematically, the number of coauthored publications is the output of a pairwise Poisson process, parameterized by the sum of the latent individual productivities $\lambda_i$ and $\lambda_j$ for coauthor pair $(i,j)$. Hence, the model parameter $\lambda_i$ gives the expected number of publications per year for author $i$, and for an author pair $(i,j)$, their joint productivity is a random variable of the form

$$P(N_{ij},t_{ij}|\lambda_i,\lambda_j) = \frac{\exp^{-(\lambda_i+\lambda_j)t_{ij}}[(\lambda_i+\lambda_j)t_{ij}]^{N_{ij}}}{N_{ij}!}, \quad (1)$$

where $N_{ij}$ is the observed number of papers coauthored by authors $i$ and $j$ over a total collaboration time period $t_{ij}$ (see Methods).

Similarly, we model prominence, defined as the number of high-impact publications, as a joint function of individual latent parameters (Fig. 1a). Mathematically, researcher prominence is modeled by a

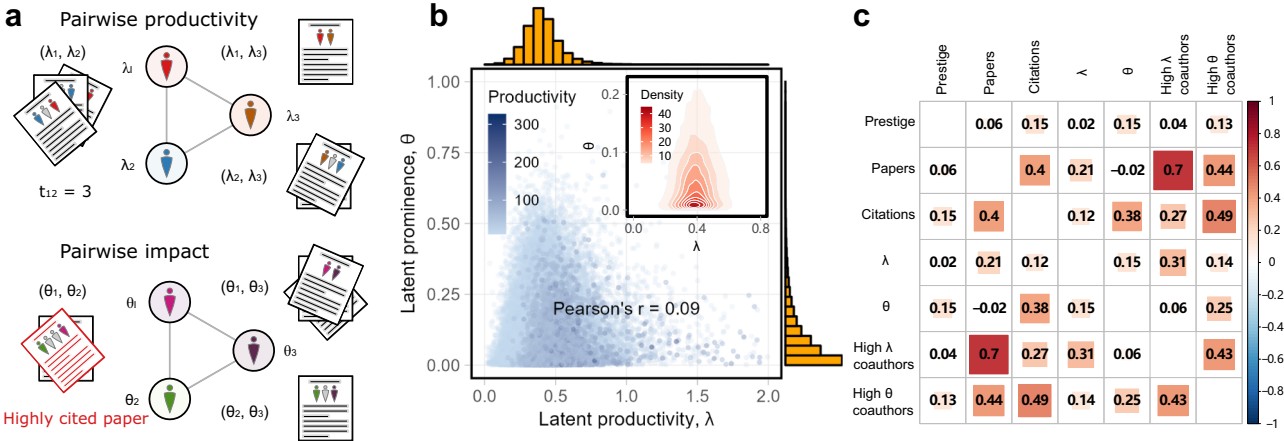

**Fig. 1 | Network decomposition and correlation of individual productivity and prominence measures. a** Illustrations of how observed individual productivity (upper) and prominence (lower) are network measures, resulting from the joint effect of the individual productivity $\lambda$ and prominence $\theta$ parameters of coauthors. **b** Joint and marginal distributions of estimated latent variables $\lambda$ and $\theta$ estimated from 198,202 mid-career STEM researchers who published at least ten papers. To better illustrate the estimated distribution of $\theta$, we omit points for 116,223

researchers with negligible $\hat{\theta} < 10^{-3}$ values. The remaining researchers have a mean $\hat{\lambda}$ value of $\mu_\lambda = 0.42$, moderately higher than the entire cohort of selected researchers reported in the main text (two-sided $t$-test, $t = 49.8$, $p < 10^{-3}$). **c** Correlation matrix of individual mid-career researchers' observed and modeled scholarly statistics, illustrating how the modeled parameters capture the network effects of collaboration.

Binomial distribution, parameterized by the sum of the latent individual prominences $\theta_i$ and $\theta_j$ of the coauthor pair $(i,j)$. Hence, the model parameter $\theta_i$ gives the expected fraction of publications with $i$ as an author that will be highly cited, and for an author pair $(i,j)$, their joint prominence is a random variable of the form

$$P(N_{ij}, m_{ij} | \theta_i, \theta_j) = \binom{N_{ij}}{m_{ij}} (\theta_i + \theta_j)^{m_{ij}} [1 - (\theta_i + \theta_j)]^{N_{ij} - m_{ij}}, \quad (2)$$

where $m_{ij}$ is the observed number of highly cited papers coauthored by authors $i$ and $j$ over a total collaboration time period $t_{ij}$ (see Methods). We note that both models assume conditional independence across publications, which may obscure some interesting temporal effects[50]. Applying these joint productivity and prominence models to all pairs of coauthors in a collaboration network yields joint likelihood functions whose independent maximization yields a set of individual productivity and prominence parameters that effectively control for the network effects of coauthorship on the variables of interest

$$L(\boldsymbol{\lambda}) = \sum_{i \neq j} \log P(N_{ij}, t_{ij} | \lambda_i, \lambda_j) \qquad L(\boldsymbol{\theta}) = \sum_{i \neq j} \log P(N_{ij}, m_{ij} | \theta_i, \theta_j). \quad (3)$$

Applied to our full dataset of 198,202 mid-career researchers across six STEM fields, defined as researchers with at least 15 years of scholarly publishing activity (see Supplementary Information), we find compelling evidence that these latent parameter models yield a useful individual decomposition of the observed joint productivities and prominences of collaborating scientists (Fig. 1b and Supplementary Fig. 3 for individual fields). Examining the marginal distributions, we find that the latent productivity and prominence variables are nearly orthogonal (Pearson's $r = 0.09$, $p < 10^{-3}$), with $\lambda$ following a Normal distribution and $\theta$ following a heavy-tailed distribution. That is, controlling for network effects, we find that individual productivity of mid-career researchers is low variance and concentrated around a central tendency of $\mu_\lambda = 0.39$ first/last-authored papers per year (standard deviation $\sigma_\lambda = 0.15$), with only the top 0.02% of researchers exhibiting a latent productivity of $\hat{\lambda} > 2$ first/last-authored papers per year.

In contrast, controlling for network effects, individual prominence is highly variable, with an average prominence of $\mu_\theta = 0.04$ (on average, for publications written by two authors, 1 out of 12.5 will be

highly cited), but a standard deviation twice as large ($\sigma_\theta = 0.08$). That is, a large majority of researchers have low individual prominence, while a minority generate a long tail of much greater impact, much like measures of popularity and wealth in other complex social systems[51]. Furthermore, both of these estimated parameters have low correlation with a researcher's career-wise raw productivity, with the Pearson correlation coefficients $r_{\lambda,N} = 0.21$ and $r_{\theta,N} = -0.02$. This implies that after controlling for the network effects of collaboration, the latent parameters could indicate the productivity and prominence of individual researchers in a given unit time period. As a technical aside, we note that parameter estimates for these models are more stable for researchers with at least 10 papers, and appear to underestimate latent productivity $\lambda$ and overestimate prominence $\theta$ for less productive authors (Supplementary Fig. 5). The distribution of $\theta$ does not qualitatively change when we alter the threshold of highly-cited papers (Supplementary Fig. 6).

If the estimated individual productivity and prominence parameters $\lambda$ and $\theta$ are genuinely measuring individual-level characteristics, controlling for network effects from collaboration, then they should only loosely correlate with their corresponding network-confounded measures of raw productivity and raw prominence. We evaluate the efficacy of these two measures by characterizing their correlation with other "unadjusted" measures and time-related dynamics for individual researchers. We first select a cohort of minimally productive mid-career researchers who have published at least 10 papers by their 15th year, and tabulate a correlation matrix of estimated individual parameters and observed scholarly statistics, based on their publications through their mid-career (Fig. 1c). We define a researcher to be "high $\lambda$" or "high $\theta$" if their individual estimated parameter is in the upper 10th percentile of same-field researchers for a given year. And, we define a high $\lambda$ or $\theta$ coauthor as a collaborator who is themselves a high $\lambda$ or $\theta$ author and has published at least three papers by the year of relevant collaboration. This correlation analysis reveals that a researcher's individual $\lambda$ and $\theta$ values correlate only moderately with their "unadjusted" productivity and prominence ($\lambda$ with papers, Pearson's $r = 0.21$; $\theta$ with citations, Pearson's $r = 0.36$), indicating that the model parameters are capturing behavior above and beyond what the unadjusted counts provide. And, we find strong evidence of the network effects of collaborations in driving the observed productivity and prominence of individual researchers,

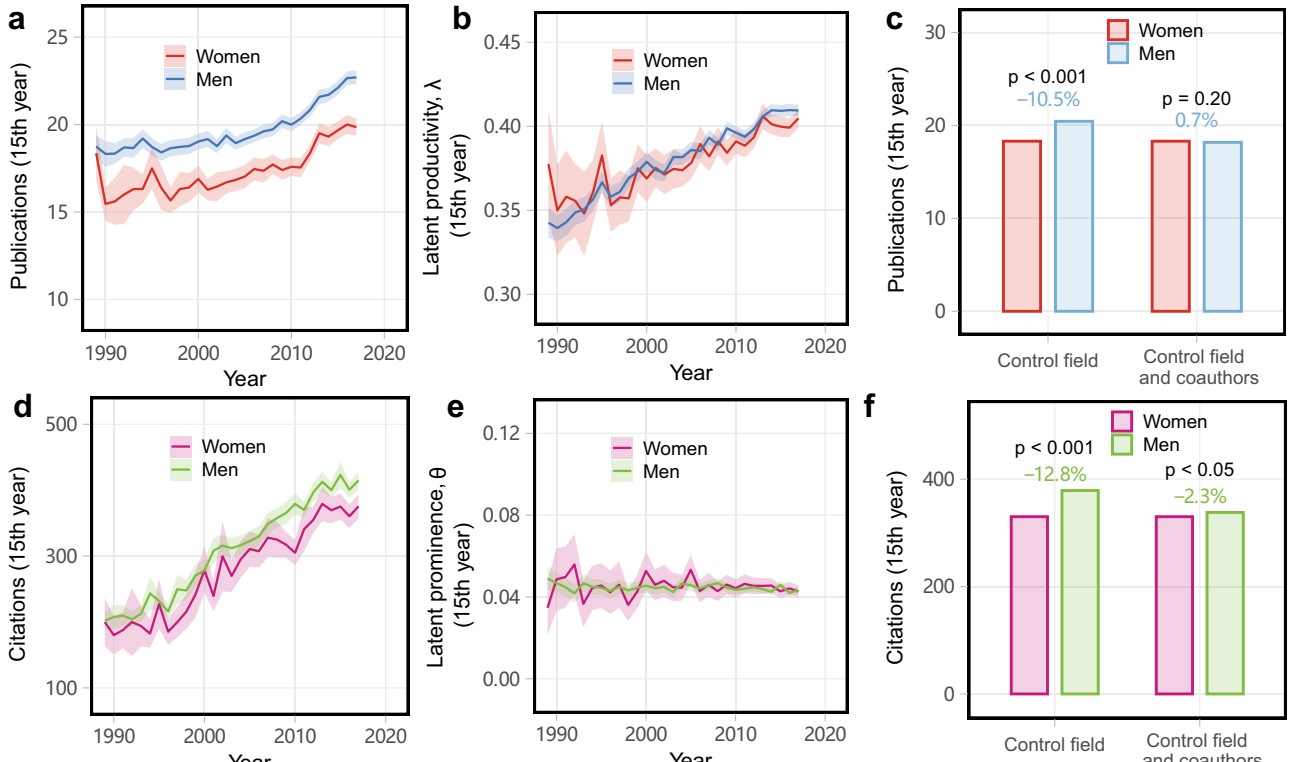

**Fig. 2 | Gendered disparities in individual productivity and prominence measures.** Across six STEM fields, observed average (**a**) productivity and (**d**) prominence, showing substantial and stable gaps, among 198,202 mid-career researchers, by gender from 1989 to 2017, along with corresponding estimated individual latent (**b**) productivity $\lambda$ and (**e**) prominence $\theta$ for the same researchers, showing negligible gendered differences. Shaded areas represent 95% confidence intervals. Then, (**c**) productivity and (**f**) prominence for pairs of men and women researchers matched on institutional prestige, year of first publication, and either (i) field alone or (ii) field and the number of coauthors, showing that gendered collaboration rates can explain the observed gendered differences in scholarly metrics. Two-sided $t$-test for comparisons.

because the number of high $\lambda$ and high $\theta$ coauthors correlates more strongly with individual productivity and prominence (papers vs. high $\lambda$ coauthors, Pearson's $r = 0.70$; citations vs. high $\theta$ coauthors, Pearson's $r = 0.49$) than do the individual's own model parameters. Hence, these network models can shed new light on the substantial but often hidden role that social networks can play in determining individual career metrics.

Similarly, if the estimated individual latent parameters are measuring a researcher's underlying characteristics, they should remain relatively stable over an individual's career path, even as their collaboration network evolves. Compared to a fully randomized null model, we find that high $\lambda$ or high $\theta$ researchers are more likely to remain in the same percentile group after 10 years (see Supplementary Information, and Supplemenatry Figs. 7–9). Furthermore, researchers with high latent parameter values in their early-career (first 5 years of publishing) are also more likely by their mid-career to be in the upper 5th percentile of citations among researchers who publish in a given field in a given year. And, this pattern holds when we repeat the analyses in matched-pair experiments, in which we match researchers on their institutional prestige, productivity, and prominence in their early-career (Supplementary Fig. 10, Supplementary Tables 1–4). These results indicate that an individual researcher's estimated model parameters for productivity and prominence are relatively stable over a career, suggesting that they are capturing underlying scholarly behavior independent of changes in collaboration patterns over time, as intended.

In agreement with past studies, we find gendered inequalities in observed measures of both career-wise productivity (Fig. 2a) and prominence (Fig. 2d) among mid-career STEM researchers, in which men both publish more papers and receive more citations than

women[22,52,53]. On average, men in these fields publish a total of 20.3 papers by the time they reach their mid-career (first 15 years) compared to 18.3 papers by women ($t$-test, $t = 24.5$, $p < 0.001$, Cohen's $d = 0.15 \pm 0.01$), and, on average, men's past publications receive 346.0 total citations compared to 330.1 citations for women's ($t$-test, $t = 4.9$, $p < 0.001$, Cohen's $d = 0.03 \pm 0.01$). In other words, men's average total productivity is 11.0% greater and they receive 5.0% more citations than women by mid-career, and these disparities are stable over time. For researchers with at least three publications in the first 5 years of their publishing career, i.e., in their early career, the probability of persisting until mid-career is 20.6% for men but only 15.7% for women, in agreement with the well-known higher drop-out rate for early-career female scientists[53]. Despite these differences in observed scholarly metrics, controlling for collaboration via our network models reveals a different pattern: across fields, the average mid-career latent productivity parameter is $\hat{\lambda} = 0.39$ for both men and women ($t$-test, $t = 0.7$, $p = 0.51$, Cohen's $d < 0.01$), and the average mid-career latent prominence parameter $\hat{\theta} = 0.044$ for men and 0.045 for women ($t$-test, $t = 0.82$, $p = 0.41$, Cohen's $d < 0.01$). That is, men and women exhibit statistically indistinguishable individual latent productivities and latent prominences, implying that the differences in observed scholarly metrics are likely caused by gendered differences in the structure and composition of researcher collaboration networks (Fig. 2b, e).

Furthermore, we find that the gendered gaps for mid-career researchers can be largely explained by variation in the number of direct coauthors in their collaboration networks. Matching women and men researchers by institutional prestige, year of first publication, and field, we still find a gendered disparity in which women's productivity and prominence is lower relative to matched men (Fig. 2c, f). However, additionally matching on the number of coauthors largely eliminates

these gendered disparities in both productivity (10.5%, $t$-test, $t = 24.5$, $p < 0.001$, Cohen's $d = 0.15 \pm 0.01$ vs. 0.7%, $t$-test, $t = 1.3$, $p = 0.20$, Cohen's $d = 0.01 \pm 0.01$) and prominence (12.8%, $t$-test, $t = 4.9$, $p < 0.001$, Cohen's $d = 0.03 \pm 0.01$ vs. 2.3%, $t$-test, $t = 2.0$, $p = 0.04$, Cohen's $d = 0.02 \pm 0.01$). Hence, we find substantial evidence that the well-known gendered productivity and prominence inequalities among women and men researchers can be largely explained as a network effect, in which the composition and size of local collaboration networks differ between men and women, and these differences lead to the observed differences in scholarly metrics, rather than any inherent difference in the researchers themselves. We note that this analysis does not establish a causal relationship, and hence known causal factors, such as the gendered impact of parenthood on researchers that leads to productivity penalty for mothers as they undertake more childcare duties[54], likely influence both productivity and collaboration networks. We also test the robustness of our findings by selecting mid-career researchers with at least 20 publications (Supplementary Fig. 13) and repeating the analysis by randomly sampling a tertile of researchers (Supplementary Fig. 14), showing that these different choices do not change the qualitative nature of our conclusions. Overall, these results suggest that collaboration networks can be viewed as a form of social capital that is distributed in unequal and gendered ways in STEM, which mediates or shapes the amount of scholarly contributions and their visibility.

If a researcher's collaboration network acts like a form of social capital, we should expect key dynamics of social capital apply in collaboration networks as well. For instance, an author's collaboration network capital should be "transferrable" to some degree between researchers. For example, collaboration by an early-career researcher with a high $\lambda$ or high $\theta$ senior coauthor should enhance the junior researcher's productivity or prominence in a way that persists into their own mid-career, compared to similar researchers without such a collaboration. For this analysis of junior-senior collaborations, we select pairs in which, at the time of collaboration, the early-career researcher is 5 or fewer years since their first publication, and the senior coauthor is 6 or more years since their first publication. Because the model estimates of individual latent parameters are more accurate for researchers with more papers, we restrict our analysis here to early-career coauthors and their senior coauthors that have at least three papers by the time of collaboration.

We find that early-career researchers are significantly more likely to collaborate with high $\lambda$ or $\theta$ senior researchers if they are based at elite institutions, which we define as research institutions whose authoritative ranking is among the top 10 in a given field (see Methods), indicating that the composition of collaboration networks itself varies with environmental prestige[55]. This may be largely due to a selection effect that high $\lambda$ or $\theta$ senior researchers are more likely to work at elite institutions, reflecting inequalities of having access to important social networks among early-career researchers. In particular, at pairwise coauthorships, the probability that an early-career researcher collaborates with a high $\lambda$ (productivity) senior researcher is 0.177 at elite institutions vs. 0.145 at non-elite institutions ($t$-test, $t = 19.3$, $p < 0.001$, Cohen's $d = 0.09 \pm 0.01$), and the probability of collaborating with a high $\theta$ (prominence) senior researcher is 0.141 at elite institutions vs. 0.067 at non-elite institutions ($t$-test, $t = 50.2$, $p < 0.001$, Cohen's $d = 0.28 \pm 0.01$).

However, regardless of the institution, researchers who collaborated with high $\lambda$ or high $\theta$ senior coauthors early in their career are significantly more likely to themselves be a highly prominent researcher in their mid-career, who have accrued the upper 5th percentile of citations among all active researchers in a given year and field (Fig. 3a, c). In particular, collaborating with at least one high $\lambda$ senior coauthor in the first 5 years of a researcher's career increases the probability of subsequently being a highly prominent researcher in the 15th career year from 16.2 to 29.5% ($t$-test, $t = 65.0$, $p < 0.001$, Cohen's

$d = 0.34 \pm 0.01$; Fig. 3a). And, a high $\theta$ senior coauthor doubles that mid-career probability from 16.3 to 39.8% ($t$-test, $t = 81.6$, $p < 0.001$, Cohen's $d = 0.61 \pm 0.01$; Fig. 3c). For both types of collaboration patterns, junior researchers from elite institutions exhibit higher productivity and prominence in the mid-career than do peers at less prestigious institutions—a disparity that reflects the value of prestigious environments[55]. This institution-based gap is larger for early-career researchers that have collaborated with high $\theta$ coauthors than with high $\lambda$ coauthors.

However, the early-career benefits of a high $\lambda$ or high $\theta$ senior coauthor appear to decrease modestly with that coauthor's career age (Fig. 3b, d). This finding contrasts with past studies of scientific mentorship[56,57], which have typically relied on unadjusted citation counts that are naturally larger for more senior collaborators and which represent a stronger confounding network effect. By correcting for the network effect of collaboration, we find instead that the benefits of collaborating with highly productive or highly prominent senior coauthors do not increase with coauthor seniority. Rather, they decrease with career age of the senior coauthor, and decrease more for high $\lambda$ coauthors, suggesting that the transfer of social capital from senior to junior researchers through collaboration is more effective earlier in the career of senior coauthors. We also test the robustness of our results by selecting senior collaborators with at least six publications and at least ten publishing career years by the time of relevant collaboration, (Supplementary Fig. 15), and we find that the different thresholds do not qualitatively change our findings.

Finally, we consider the impact of environmental prestige on latent productivity and prominence of mid-career researchers. Past work has shown that working at a more prestigious institution drives greater productivity and prominence among early-career researchers[55]. However, as with past work on the impact of mentorship, such insights were derived from scholarly measures that did not control for the network effects of collaboration, which increase as a career progresses. Across six STEM fields, researchers in our dataset affiliated with elite institutions on average publish a total of 21.8 papers up to their mid-career (first 15 years), which is 8.5% greater than the 20.1 for researchers at non-elite institutions ($t$-test, $t = 11.5$, $p < 0.001$, Cohen's $d = 0.11 \pm 0.02$, Fig. 4a). And, over the same career time, researchers at elite institutions receive on average 493.7 citations, which is 62.1% greater than the 304.5 citations received by researchers at non-elite institutions ($t$-test, $t = 27.8$, $p < 0.001$, Cohen's $d = 0.38 \pm 0.02$, Fig. 4d). Hence, in unadjusted scholarly metrics, researchers at elite institutions have marginally higher productivity and a substantially higher impact.

We find that these productivity and prominence advantages for researchers working in prestigious environments also appear in our estimated individual latent parameters. Researchers at elite institutions, on average, also exhibit a marginally greater latent productivity than those at non-elite institutions ($\lambda = 0.394$ vs. 0.387; 1.8% greater; $t$-test, $t = 6.0$, $p < 0.001$, Cohen's $d = 0.05 \pm 0.02$, Fig. 4b). And, these same researchers, on average, exhibit nearly double the latent prominence of researchers at non-elite institutions ($\theta = 0.071$ vs. 0.037; 91.9% greater; $t$-test, $t = 36.7$, $p < 0.001$, Cohen's $d = 0.43 \pm 0.02$, Fig. 4d). Hence, controlling for the network effects of collaboration, we find smaller but still significant advantages in productivity but even larger advantages in prominence for researchers working at elite institutions, compared with raw scholarly metrics. The persistence of the advantages of elite environments after controlling for network effects suggests that other factors likely drive these differences[55], e.g., differences in resources, the size of collaboration networks, or selection effects that apply primarily to mid-career researchers. In addition, we find that the results do not qualitatively change when we modify the number of selected elite institutions to the top 20 (Supplementary Fig. 16).

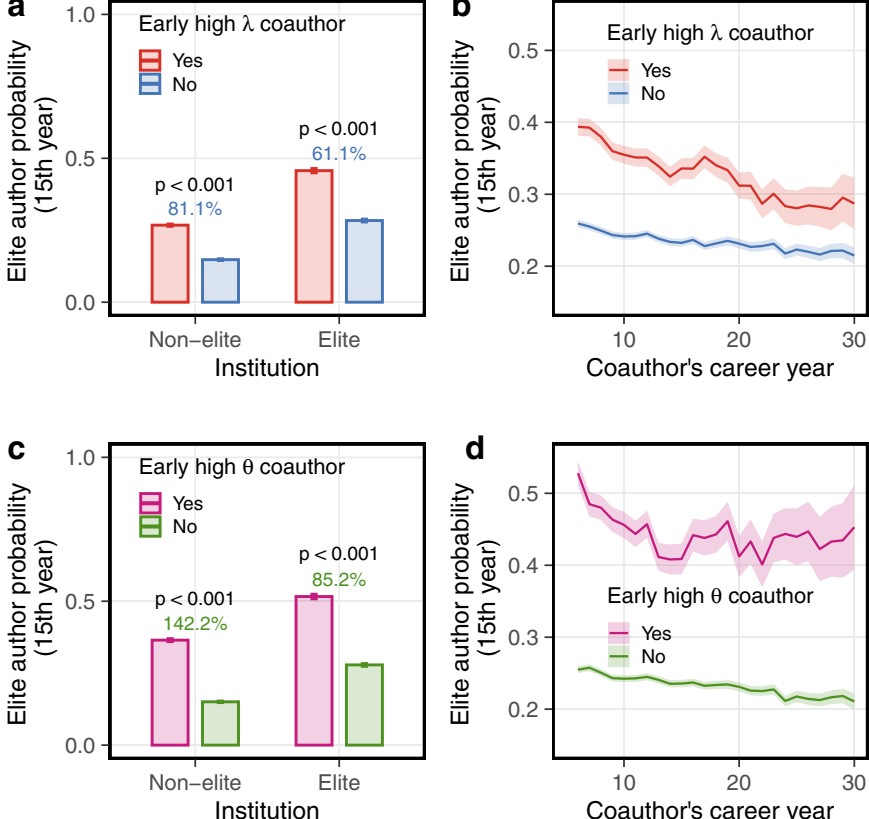

**Fig. 3 | Impact of senior coauthors on junior researcher's likely mid-career impact.** Early career researchers with a (**a**) high $\lambda$ or (**c**) high $\theta$ senior coauthors are substantially more likely to be elite authors by their mid-careers, regardless of institutional prestige. The magnitude of this effect for both (**b**) $\lambda$ and (**d**) $\theta$ is large regardless of the career age of the senior coauthor, but decreases modestly with

age. Junior researchers with early high $\lambda$ coauthors $n$ for yes = 57,552; no = 229,225. Junior researchers with early high $\theta$ coauthors $n$ for yes = 30,983; no = 255,794. Two-sided $t$-test for comparisons. Error bars in (**a** and **c**) indicate mean ± 1.96 SEM. In (**b**, **d**), solid lines indicate mean and shaded areas indicate 95% confidence intervals.

Some of this prestige advantage can be explained by differences in the composition of a mid-career researcher's collaboration networks. Matching researchers in our sample by field and year of first publication, we find that researchers at non-elite institutions are only 6.8% less productive than those at elite institutions (Fig. 4c). However, further matching on variables that quantify the composition of a researcher's collaboration network, and in particular, the number of coauthors, number of high $\lambda$ coauthors, and number of high $\theta$ coauthors, we find that researchers at non-elite institutions are 2.8% more productive than those at elite institutions ($t$-test, $t = 3.1$, $p < 0.01$, Cohen's $d = 0.04 \pm 0.02$). These network effects are even stronger for the prominence of individual researchers. Matching researchers by field and year of first publication, researchers at non-elite institutions receive 39.9% fewer citations than those at elite institutions, while further matching on collaboration network variables shrinks this gap to only 19.9%. Hence, in contrast to gendered differences (Fig. 2), we find that the inequalities in productivity and prominence associated with environmental prestige cannot be explained entirely by differences in the structure of collaboration networks, suggesting that additional prestige-related variables play an important role in driving the greater scholarly impact of researchers at elite institutions.

In addition, we test the interaction effects of gender and institutional prestige on the performance of mid-career researchers. We find that the prestige of institutions has a relatively stronger effect on researchers' productivity and prominence than gender, for both unadjusted measures and latent parameters (see Supplementary Fig. 12). In particular, both gender and institutional prestige have negligible effects on latent productivity $\lambda$, while institutions appear to

have stronger influence than gender on latent prominence $\theta$. The observation that prestige does not appear to drive latent productivity $\lambda$ is supported by other recent studies, which show how the greater productivity of faculty at prestigious departments can be largely explained by a collaboration network effect: elite departments provide more available funded research labor, who then coauthor papers with the faculty members in their departments[58].

## Discussion

By mediating scientific attention, evaluation, and collaboration, social networks play a fundamental role in shaping both the advancement of science and the pervasive social and epistemic inequalities that appear in most scientific communities. However, analyses of scholarly metrics associated with productivity and prominence, based on counts of publications and citations, even when normalized in some way, as in the case of fractional authorship or adjusting for journal impact factors, tend to be confounded by network effects that operate above the level of individual publications. Such network effects make it difficult to gain insight into the causes and consequences of these inequalities, particularly across the span of a scientific career. Here, we introduced two scholar-level generative network models that allow us to estimate parameters that represent individual researcher productivity and prominence, while controlling for the effects of collaborations with more or less productive or prominent collaborators over time (and those collaborators' collaborations, etc). We then applied these models to a large dataset of 198,202 mid-career researchers and all of their first-last author collaborations across 70 years of time and six fields in STEM to investigate the effect of collaboration networks.

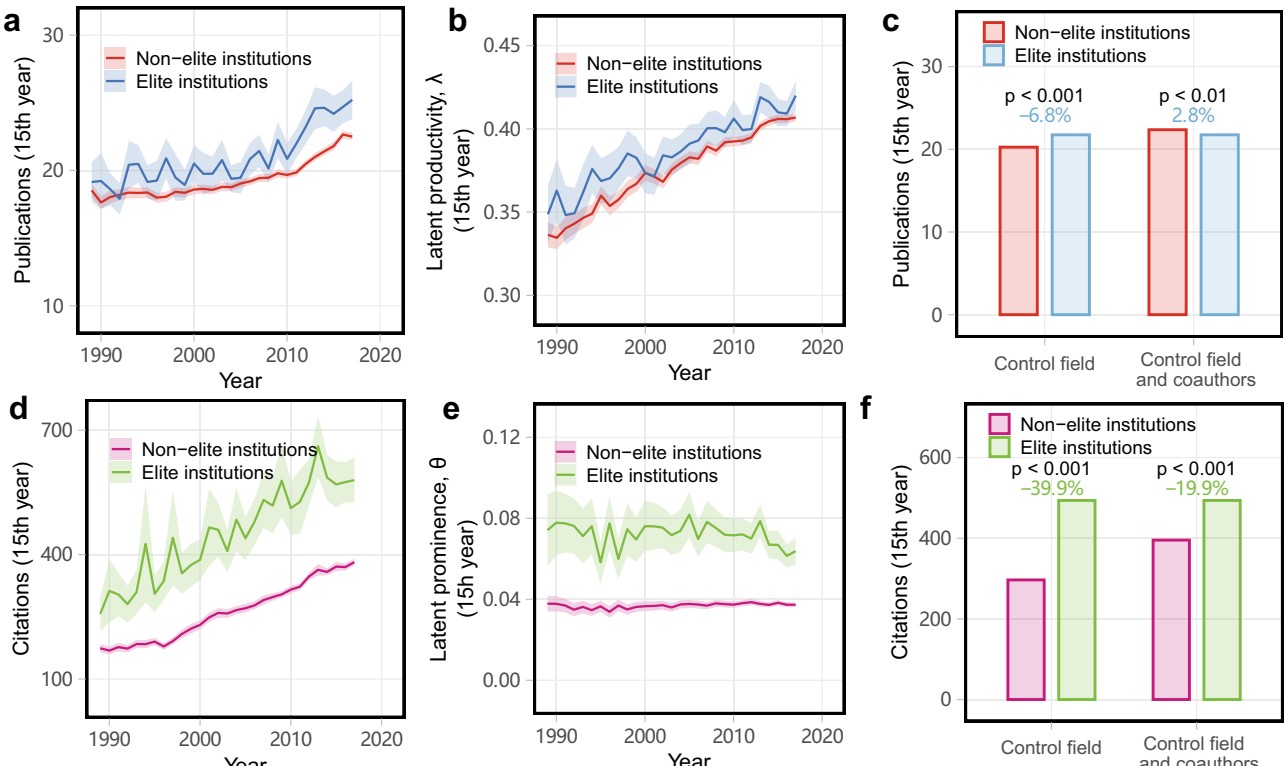

**Fig. 4 | Impact of elite environments on researcher productivity and prominence.** The institution-based differences in unadjusted average (**a**) productivity and (**d**) impact, and in latent variables (**b**) $\lambda$ and (**e**) $\theta$ of mid-career researchers. Shaded areas indicate 95% confidence intervals. Then, (**c**) productivity and (**f**) prominence for pairs of mid-career researchers matched on institutional prestige, year of first publication, and either (i) field alone, or (ii) field, the number of coauthors, the number of high $\lambda$ coauthors, and the number of high $\theta$ coauthors, showing that elite working environments can explain the observed differences. Two-sided $t$-test for comparisons.

We find that the observed gendered gap in productivity and prominence can be largely explained by differences in social networks. The way social networks can behave like social capital, with boosting effects on junior researchers decaying as their senior collaborators age. After controlling for network effects, our adjusted productivity and prominence parameters can explain a significant proportion but not all scholarly disparity related to environmental prestige. These results have implications for gendered and institutional differences in scholarship, which we discuss further in the following paragraphs.

Our estimated latent parameters reveal that women researchers who persist until mid-career (15 years since first publication) exhibit equal productivity and prominence to persisting men (Fig. 2). This finding suggests that the well-known gendered difference in "unadjusted" scholarly metrics like number of papers (productivity) and total citation counts (impact) can be explained by gendered differences in coauthorship networks. Although this result does not imply causality, it does indicate that known causal factors like the gendered impact of parenthood on researchers[54], likely also shape collaboration networks. By providing new individual parameters after adjusting network effects, our findings highlight the importance of social networks in shaping scholarly gender differences among mid-career researchers, which contributes to the abundant literature on potential causes and effects of gender disparity in science, including academic culture[19] and homophily[15,16]. More research is needed to identify the likely multiple reasons that women on average have fewer coauthors than men, and the degree to which those reasons relate to scholarly factors, preferences, or non-meritocratic factors.

These results also suggest that collaboration networks can be viewed as a form of social capital that is distributed in unequal and gendered ways in STEM. In this way, collaboration networks may serve as a common mediating variable for other social and epistemic

inequalities, which may then drive differences in the amount or visibility of scholarly contributions, or other factors associated with scientific discovery. Efforts specifically aimed at expanding and supporting the collaboration networks of women researchers, e.g., formal support and advocacy organizations, women-in-science meetings, and fellowships for women that support intensive new collaborations, seem likely to help mitigate these gendered gaps in scholarly metrics, and to broadly support scientific discovery.

Supporting the view that collaboration networks act like a form of social capital, we find that early-career collaborations with elite senior researchers, as identified via their high latent parameters $\lambda$ or $\theta$, seem to raise the latent productivity and prominence of their junior coauthors, which supports the long-term development of their academic careers (Fig. 3). This effect appears regardless of the prestige of the affiliated institution, but is amplified in prestigious environments, which measurably catalyze the formation of collaboration ties with elite researchers. However, the boosting effect that early-career collaborations with elite senior coauthors have on mid-career productivity and prominence gradually declines as senior coauthors age, regardless of the senior authors' latent parameter values. Further research is needed to understand the causal mechanisms through which these senior collaborations produce lasting influence on the productivity and prominence of early-career researchers, whether these effects are gendered, and what causes the age-related effect.

Many possibilities are plausible. The effect could reflect epistemic ossification, in which older scientist become progressively less well-connected to the dynamic core of their field. It could also reflect social saturation, in which the capacity of senior scientists' collaborators to from new collaborations with junior colleagues is gradually depleted. A particularly plausible possibility is that the effects are driven by prestige-correlated selection and social stratification. For instance,

elite senior researchers are more likely to be based at prestigious, research-intensive institutions, and hence are more available to collaborate with students intent on pursuing academic research careers, who have enhanced prospects to do so, as a result of their prestigious pedigree. By the same token, talented students at a less prestigious institution will have fewer available elite researchers to collaborate with, and hence have lower access to the kinds of social capital that facilitate a successful early research career. Or, the advantage of mid-career researchers at elite institutions in their productivity and prominence may reflect the stratification of research resources, e.g., funding, research group size, computational or experimental facilities, etc., and early collaborations with elite senior researchers simply increases the likelihood of ultimately working at such an institution. Identifying the underlying causes of the long-term effects of these collaborations is an important direction of future research, with specific implications for efforts to mitigate social and epistemic inequalities in science.

Overall, our findings shed considerable new light on the fundamental role of collaboration networks in shaping scientific careers and mediating scholarly inequalities. Our results suggest that collaboration networks embody a form of unequally distributed social capital, which influences who makes what scientific and technological discoveries. In particular, collaboration network effects can explain both the persistent gendered inequalities among mid-career researchers in productivity and prominence, and a considerable portion of the observed inequalities between researchers working in more or less elite environments. While these results are not causal, they do suggest that a more detailed understanding of the factors that influence the size and composition of researcher collaboration networks is likely to bring us closer to a causal understanding of many social and epistemic inequalities in science. Collaboration networks may also play an important role in the domain of research and development efforts, particularly in the form of patent collaborations[59]. Studies focusing on cross-disciplinary effects thus are likely to shed further light on the dynamics and influence of social capital in scientific discovery, and the role of collaboration networks in shaping individual research careers.

There are several limitations to our analyses. By focusing only on first and last author collaborations, we neglect all collaborations with middle coauthors, regardless of the kind or size of their contributions. This categorical selection mitigates the confounding network effects of large author lists, but also neglects the value and influence of team science. Among the six STEM fields studied here, a common norm is that research tasks like data analysis, experiments, and visualization are performed by the first author, while the last author commonly plays the more supervisory role of research design, manuscript writing, and funding support. The specific and varied roles of and interactions with middle authors are omitted in order to simplify the model framework. Elaborating our modeling framework to incorporate the effects of middle-author collaborations, perhaps labeled using an author contribution taxonomy, may reveal additional nuance or secondary effects of interest. In addition, in order to produce reliable estimates of latent parameters, researchers with only a small number of collaborations were dropped from our analysis, which limits our insights to relatively productive mid-career researchers. Hence, we can say little about the degree to which our results hold for researchers with short track records. Our name-based gender classification used data from the US Social Security Administration, which is biased toward English names. Further studies that focus on gender disparity of other ethnic groups are needed to show if similar gendered network patterns persist. And, our analysis of environmental prestige used only a coarse dichotomous variable for elite or non-elite institution, which likely obscures the effects of gradations of prestige. Finally, our analyses depend on crude but easy-to-measure metrics of scholarly contributions, based on publication and citation counts, which can be useful in aggregate but should not be confused with measures of scientific utility.

Our results implicate a fundamental but complicated role for collaboration networks, and the kind of social capital they embody, in forming and perpetuating social and epistemic inequalities in the scientific processes of STEM fields. They also suggest that collaboration network effects could be leveraged to help mitigate some of those same inequalities, to better support scientific discovery and to broaden participation in science. For instance, targeted support of cross-institution, early-career collaborations with elite senior researchers, perhaps through specialized fellowships, may support the career advancement of promising young researchers who would otherwise leave research. Similarly, directly supporting the collaboration networks of women researchers may improve both retention and productivity, particularly at times when gendered impacts occur, e.g., at parenthood[54]. And, efforts to "correct" for collaboration network effects when evaluating candidates for faculty positions or applicants for funding is likely to help mitigate the multiple implicit biases that are known to favor elite-pedigree men researchers with prolific senior collaborators[1,5,25]. Network effects are a natural part of the social processes that underlie the scientific process, and are likely to be key components in any effort to mitigate social and epistemic biases, to make academia more meritocratic and less sensitive to the effects of cumulative advantages.

We note that our models are a general way to decompose observed data on repeated collaborative activities, such as technological inventions, business partnerships, and musical composition, into individual contributions. Applying similar models to other phenomena would be an interesting direction of future work, which may help illuminate individual differences and contributions to these group activities. As we have done in this paper, it can also shed new light on how those differences relate to other variables of interest and, in particular, the role of those differences in driving broader social inequalities.

## Methods
### Publication and citation data
We use the MAG dataset, containing journal articles and conference proceedings published between 1950 and 2019, inclusive. MAG provides a 5-level taxonomy of academic fields of study; the top level 0 divides all documents into 19 major fields. Among them, we select six scientific fields representative of the traditional science, technology, engineering and mathematics (STEM) domains: biology, chemistry, computer science, mathematics, medicine, and physics. These fields publish the majority of research papers in science and technology domains (see Supplementary Fig. 1). Following the publication norms in these fields, we include only journal articles in our analyses for all fields except computer science. For computer science, where conference proceedings are peer reviewed in the same way that journal articles are in other fields, we include both journal and conference articles.

Missing researcher affiliations are common in MAG, but difficult to impute. The MAG dataset includes 80.4 million papers that meet the above inclusion criteria. Among these, 36.0 million papers provide author affiliation information, and we consider only these in our analyses. These affiliations provide necessary information for assessing the environmental effects on coauthorship, career development, productivity, etc. for individual scientists. The reasons for missing affiliations for authors in MAG remain unclear.

Our analyses consider coauthorship only between first and last authors of each paper. In the six STEM fields we analyze, the first and last authorship positions are typically understood to denote the authors that made the greatest contributions to the research. There are circumstances where this norm does not apply, e.g., in specific subfields where authors are listed in the alphabetical order, or when there are multiple "first" or "last" authors due to equal contribution flags, as well as in some large collaborations. To account for this latter

category, we exclude all papers with more than 10 listed authors. Applied together, our refined dataset contains 12.9 million unique authors and 20.0 million research articles. Our first-last author counting scheme eliminates the effects of large author lists and the relevance of fractional counting, at the expense of potentially under-counting contributions and effects of middle-authorship. Most authors are associated with very few publications, and our analyses focus on the mid-career trajectories of the 198,202 productive authors that published their first paper in 1975–2003 and have at least 10 publications in the 15th career year.

We define the highly cited papers to be those that receive the upper 8th percentile of citations among papers published in journals and computer science conferences, respectively, for a given year and level 0 field annotated in MAG dataset. In MAG, a paper belongs to exactly one level 0 field but falls into several different fields at other fine-grained levels, making it difficult to operationalize the definition of highly-cited works based on these levels. The theoretical need to normalize citation counts at a fine-grained level only applies when authors are being compared directly across such fields, and that our models naturally account for such cross-field variability as our model essentially estimates a researcher-specific parameter.

### Institutional prestige and elite institutions
For a specific discipline, we use the $z$-score of the number of total historical highly cited papers produced by each research institution to define its prestige score

$$p_i^{\mathrm{inst}} = \frac{N_i^{\mathrm{high}} - \langle N^{\mathrm{high}} \rangle}{\sigma / \sqrt{n^{\mathrm{inst}}}}, \qquad (4)$$

where $N_i^{\mathrm{high}}$ is the number of highly cited papers produced by institution $i$, $\langle N^{\mathrm{high}} \rangle$ is the average number of highly cited papers by all institutions, $\sigma$ is the standard deviation of highly cited papers, and $n^{\mathrm{inst}}$ is the number of institutions. The institutional prestige score is discipline specific, but does not vary over time. We define the top 10 research institutions by this measure, within each field, to be elite institutions.

### Gender
We assign binary gender labels to authors according to a classifier based on U.S. Social Security Administration data, which records the historical gender associated with names of newborn babies in the United States of America[60]. Hence, our analysis of gender disparity is most applicable to researchers with origins in North America or other native English-speaking countries. Only first names that have at least 95% accuracy for a specific gender are retained for the matching. As such, we matched 126,805 productive authors for our analysis who published the first paper in 1975–2003, composing 64.0% of all productive mid-career authors selected for our study, among which 20.2% (25,666) are women.

### Latent variable estimation
For each network model and each field, we use all papers published within that field up to a given year, and estimate the latent parameter sets using convex optimization. We estimate yearly parameters with bootstrap-corrected pseudo-likelihood using 30 replications for every year from 1975 to 2017. For each year $T$, we construct the coauthorship network by using all publications from 1950 to $T$. In each round of bootstrap sampling, prior to estimating the network models, we prune all subgraphs of the coauthorship network that are trees, as model parameters become non-identifiable in such structures. Authors dropped as a result of pruning are assigned a latent variable of 0, and the final parameter estimates are the average values of all replications. In our analysis of patterns over time, authors receive latent parameter estimates in every year from their first appearance as either a first or last author until 2019.

Individual research latent parameters $\lambda$ and $\theta$ are estimated using the convex optimization R package $CVXR$[61]. Within a given field, for each year from 1975 to 2017, we estimate the model parameters on a bootstrap of all papers published up to and including that year, using 30 replications. An individual researcher's $\lambda$ and $\theta$ parameters are recorded as the (bootstrap) average across replications.

In our network models, we assume that a pair of coauthors started their collaboration 1 year before they published their first paper together. Hence, the duration of collaboration for authors $i$ and $j$ is $t_{ij} = \mathrm{Yr}_{ij}^{\mathrm{lastpaper}} - \mathrm{Yr}_{ij}^{\mathrm{firstpaper}} + 1$.

We assess the bias induced from pruning authors in collaboration network trees by examining the differences in individual-level attributes such as institutional prestige and gender in the 2017 network for retained and dropped authors. There are 35.6% women authors in the retained population, while the proportion of women is 31.9% in the dropped population. And, the average institutional prestige score for retained authors is 5.51, and 3.41 for dropped authors, suggesting that authors in the tree subgraphs are usually from less prestigious institutions.

### Data manipulation and visualization
We used R package data.table version 1.14.0 for processing and manipulating publication and citation data[62]. All data visualization graphics in this study are made with the R package ggplot2 version 3.3.5[63].

### Reporting summary
Further information on research design is available in the Nature Research Reporting Summary linked to this article.

## Data availability
The Microsoft Academic Graph data was obtained by following the guidelines at https://docs.microsoft.com/en-us/academic-services/graph/get-started-setup-provisioning.

## Code availability
The code used in this study has been deposited in the GitHub repository https://github.com/LleytonLi/LatentVariables.

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

## Acknowledgements

This research was supported by the High Performance Computing Platform of Beihang University. We thank H. Zheng and K. Shoub for helpful discussions. W.L. was supported in part by the Alexander von Humboldt Foundation. S.Z. was supported by a NSF Graduate Research Fellowship Award DGE 2040434. Z.Z. was supported by Program of National Natural Science Foundation of China Grant No. 11871004, 11922102, 62141605 and National Key Research and Development Program of China Grant No. 2018AAA0101100, 2021YFB2700304. S.J.C. acknowledges support from the NIH (R-34, DA043079-01A1) and NSF (SES-1514750, SES-1461493). A.C. was supported in part by Air Force Office of Scientific Research Award FA9550-19-1-0329.

## Author contributions

W.L., S.Z., Z.Z., S.J.C., and A.C. designed and performed research, contributed new analytic tools, and wrote the paper. W.L. analyzed data and made the visualizations.

## Competing interests

The authors declare no competing interests.
