## [Peer Review File · Nature Communications]

Reviewers' Comments:

Reviewer #1:

Remarks to the Author:

The work aims at assessing the network effects of research collaborations on the productivity (measured by number of publications) and prominence (measured by high impact publications) of individual scientists. To that purpose the authors apply two network models, by which they try to explain also: i) whether differences in collaboration behaviour across gender have an effect on productivity and prominence; and ii) the extent of transferability of network capital from senior to junior scientists. The authors conclude that there is a strong relation among the variables explored.

The questions posed by the authors are intriguing and significant to the field of research assessment. Unfortunately, while their network models appear quite sophisticated, they apply them on quicksand. It is evident that the authors are not familiar with the fundamentals of the microeconomic theory of production, and the basics of scientometrics, which led them to choose an invalid indicator of productivity, and incur in several methodological flaws.

I had a presentiment of that from the very beginning, when the authors stated that (page 2, line 42): "Scientists at elite institutions also receive disproportionately more funding than those at less prestigious institutions, which may enable greater scientific productivity". As a matter of fact, productivity is a ratio of output to input. To raise productivity the marginal increase of output needs to exceed the marginal increase of input.

The authors' definition of individual scientists' productivity, as the number of publications in a period of time, is unacceptable, as publications are not all worth the same. All others equal, two automobile manufacturers producing the same number of cars, one producing small utility cars and the other SUVs, do not have the same productivity. The authors should have used total impact in place of the number of publications.

Moreover, the authors apply the full counting method, instead of the fractional one. Assuming that a publication corresponds to a scientific discovery, all others equal, to each co-author one should assign a proportion of that publication, not to favour them vis-à-vis colleagues producing a publication with fewer co-authors. The number of discoveries is two, and not the sum of co-authors.

Furthermore, it seems that the authors are working at top level O of MAG field classification. If it is true, citations need to be field-normalized at finer-grained level, because the citation behaviour varies a lot within top level O fields.

The accuracy of authorship disambiguation is critical in these kinds of analysis. It is important then to provide more details on the precision and recall of the disambiguation algorithm applied.

Finally, the authors' review of the literature and the methodology section lack a large number of pertinent studies published in bibliometrics journal (e.g. *Scientometrics*, *Journal of Informetrics*, *Research Evaluation*, etc.), which confirms the scarce familiarity of the authors with scientometrics and its consequences on the soundness of the work. Only two works out of 42 in the references list are published in bibliometric journals.

Reviewer #2:

Remarks to the Author:

The research is timely and provides a novel contribution to the literature. The findings that network effects have more influence on publishing quantity and impact than other variables like gender and institutional prestige support emerging work in this area. The perspective of the literature review, and interpretation of the results, are consistent with recent research showing collaboration networks can largely explain differences in publishing rates. The current manuscript advances this argument by also looking at article impact (citation rates), beyond mere quantity of publication. There is also more refined data analysis in excluding low rate publishers and focusing on mid-career scientists and their senior collaborators. Further, the current paper controls for confounding variables.

The only area I can see improvement in is the details of the methodology. How were the publication years selected? How were the STEM fields selected? That is, why is engineering excluded but medicine included?

Reviewer #3:

Remarks to the Author:

The manuscript titled "Untangling the network effects of productivity and prominence among scientists" by Li et al. is a well written and well-documented analysis integrating collaboration networks, institutional prestige and gender information to explain the variation in observed productivity and prominence across a large set of STEM researcher profiles based upon the Microsoft Academic Graph dataset.

A key result based upon their analysis that controls for network effects, is that productivity is low-variance and that prominence is high-variance; and also the stability of the estimated productivity and prominence parameters over the career; and the statistically indistinguishable parameters by gender. In general, the manuscript exhibits robust and insightful results, a strong methodology, and strong visualization. Yet there are a few concerns regarding the true generalizability of the results as stated.

First, the choice of constructing the social network based upon first and last authors of papers requires additional justification and robustness check. One reason for this is that it means that many mid-career scientists who are commonly found in the middle positions are omitted, and so it's not clear to what extent this affects the results of the study which the authors state is aimed at identifying differences in productivity and prominence of mid-career researchers; see Zeng XH, Duch J, Sales-Pardo M, Moreira JA, Radicchi F, Ribeiro HV, Woodruff TK, Amaral LA. Differences in collaboration patterns across discipline, career stage, and gender. PLoS biology. 2016 Nov 4;14(11):e1002573. An additional pertinent analysis relating team size and annual productivity across the career are reported in Petersen AM, Riccaboni M, Stanley HE, Pammolli F. Persistence and uncertainty in the academic career. Proceedings of the National Academy of Sciences. 2012 Apr 3;109(14):5213-8. A similar confounder occurs at the data selection stage, where it is indicated that roughly only 1 in 2 MAG articles contain affiliation data, and so the authors exclude those without affiliation data. This seems like a large sample selection bias combined with the first-last author criterion, that highly conditions the results.

Second, the language regarding "network effects" is a bit of an embellishment ("coauthorship effects" may be more suitable here), as the authors are primarily capturing the first-order network (ego-network) of researchers, as opposed to the entire collaboration network which the authors convey in the abstract and introduction.

Another less central issue is the language that suggests that their method is correct and that traditional measures are incorrect: "efficacy of these two measures by characterizing their correlation with other "uncorrected" measures and time-related dynamics for individual researchers", which is a bit presumptive. Possibly this is just an issue of word choice, and so adjusted could replace corrected and ameliorate this issue. Moreover, in order to assess the efficacy in the case of theta, a better baseline for comparison would be the solo-authored papers by the same author (for which $\theta_{i,j} = 0$ by construction).

One final issue are the results comparing early-career researchers collaborating with senior researchers at elite institutions. Clearly there is a section issue that explains the result, that being that high lambda or that researchers are more likely to be at elite institutions.

In summary, despite the above concerns, this work merits strong consideration for publication as certain findings (eg regarding gender) are in very near alignment with previously reported findings, and so given that appropriate comparative approaches are used in this analysis, there is much reason to trust the differences to be robust to the underlying caveats.

Additional Comments:

- Figure 1: Panel B - A scatter plot is not the most informative plot as there are too many datapoint to appreciate the density, and so a 2-d density plot would be better suited. Panel C - diagonal elements should be removed as they render other relevant relationships difficult to

distinguish in magnitude

- Statements including "papers per year" should be rephrased as "first/last-authored papers per year"

- I found the statement "indicating that they are performing as desired in controlling for the network effects of collaboration" on line 133 odd yet perplexing, and so I believe it merits elaboration & clarification.

- There are a number of arbitrary thresholds used in the manuscript (eg "at least 3 papers by the year of relevant collaboration" and "the early-career researcher is 5 or fewer years since their first publication, and the senior coauthor is 6 or more years since their first publication.") which further raise the issue of robustness: how were these thresholds chosen and to what degree do results change if they are for example doubled?

Reviewer #4:

Remarks to the Author:

This paper consists of an innovative analysis of STEM researchers' collaboration networks and how they relate to differences in productivity and prestige depending upon gender and institutional status.

My comments below mainly focus on the extent of the contribution and the analytic approach.

1.) Contribution

The authors have discovered that gender gaps in productivity and prominence are mitigated when accounting for collaboration networks (operationalized in terms of first-author and last-author pairs). The dataset constructed and the analytic approach are quite innovative. The authors also have discovered that scientists working at elite institutions have a distinct advantage, and that this disparity is not mitigated when controlling for network effects. In fact, researchers working in a non-elite institution appear to be at a distinct disadvantage even when working with a prominent co-author.

Plenty of extant research has shown that women have less access to important social networks compared to men, and that these gender gaps in network access and brokerage help to explain gender differences in career outcomes (Belliveau, 2005; Greguletz et al., 2019; Ibarra, 1992, 1997; for a meta-analysis, see Fang et al., 2020) and in STEM in particular (Abramo et al., 2013; Bozeman & Curley, 2004; Collins & Steffem 2019; see Casad et al., 2021, section entitled "Social Capital" for a short review of gender, social capital, and STEM). Thus, although the findings reported here are compelling, it would be helpful to understand further how these findings contribute above and beyond other past work on gender and social networks and the concomitant effects on careers. In fact, I think that the dataset in this paper provides a rich portrait of collaboration networks and makes some unique contributions, especially in the STEM context. Overall, I recommend that the authors elaborate on how these findings make a unique contribution in terms of understanding the relationships among gender, social capital/networks and career outcomes in STEM.

Furthermore, the authors analyze gender gaps and institutional (elite/non-elite) gaps separately. The gaps are, in fact, much larger for institutional status versus gender. Did the authors test an interaction between gender by institutional status and parse those findings? I would be curious to see whether institutional status trumps gender. Based on the main effects of gender versus institutional status, it appears that one's institution has a much stronger effect on career outcomes compared to gender.

2.) Analytic approach

Overall, I applaud the authors for conducting such a thorough and comprehensive analysis. Their efforts are impressive. I raise the following issues/questions below to help readers gain clarity over

their sampling and analytic decisions made.

-Thresholds:

The choices for certain thresholds used in the analyses were a bit unclear. For example, why is prominence modeled as the top 8% of citations? Why not 5% or 10%, for instance? Similarly, for the elite institution threshold, why top 10 as opposed to top 5, top 15 etc.? Similarly, why did the authors choose 10 publications within 15 years for their sample? This seems like a fairly low threshold for productivity.

-Sensitivity analysis:

Related to the point above about sampling, it would be interesting to see whether the patterns found hold when examining the sample by quartiles or tertiles given the large differences in productivity among the researchers sampled.

- Effect sizes

Please report effect sizes for significant results. Given the size of the dataset, the authors report highly significant effects, but some of the absolute values of the numbers are fairly small (e.g, the means by gender of number of published papers reported on the bottom of p. 7).

Additional comments:

-Did the authors investigate whether the gendered make-up of the co-author pairs affected the results? I could not find a discussion of that in the paper.

-Please elaborate on the connection to parenthood status made at the top of p. 14. I don't follow how these results are related to work on parenthood status. Are the authors implying that women have fewer collaborators, and thus don't reap the benefits of collaboration, because they have more caregiving demands? That's interesting, but please elaborate on this point further, as well as how the variables collected relate to that point.

References

Abramo, G., D'Angelo, C. A., & Murgia, G. (2013). Gender differences in research collaboration. *Journal of Informetrics*, 7(4), 811-822.

Belliveau, M. A. (2005). Blind ambition? The effects of social networks and institutional sex composition on the job search outcomes of elite coeducational and women's college graduates. *Organization Science*, 16(2), 134-150.

Bozeman, B., & Corley, E. (2004). Scientists' collaboration strategies: Implications for scientific and technical human capital. *Research Policy*, 33(4), 599-616. <https://doi.org/10.1016/j.respol.2004.01.008>

Collins, R., & Steffen, N. (2019). Hidden patterns: Using social network analysis to track career trajectories of women STEM faculty. *Equality, Diversity and Inclusion*, 38(2), 265-282. <https://doi.org/10.1108/EDI-09-2017-0183>

Casad, B. J., Franks, J. E., Garasky, C. E., Kittleman, M. M., Roesler, A. C., Hall, D. Y., & Petzel, Z. W. (2021). Gender inequality in academia: Problems and solutions for women faculty in STEM. *Journal of Neuroscience Research*, 99(1), 13-23.

Fang, R., Zhang, Z., & Shaw, J. D. (2020). Gender and social network brokerage: A meta-analysis and field investigation. *Journal of Applied Psychology*, 106(11), 1630-1654.

Greguletz, E., Diehl, M. R., & Kreutzer, K. (2019). Why women build less effective networks than men: The role of structural exclusion and personal hesitation. *Human Relations*, 72(7), 1234-1261.

Ibarra, H. (1992). Homophily and differential returns: Sex differences in network structure and access in an advertising firm. *Administrative Science Quarterly*, 37(3), 422-447.

Ibarra, H. (1997). Paving an alternative route: Gender differences in managerial networks. *Social Psychology Quarterly*, 60(1), 91-102.

Response to the reviews of manuscript NCOMMS-21-50002: “Untangling the network effects of productivity and prominence among scientists”

Dear Editors and Reviewers,

Thank you for our detailed attention to our manuscript, from technical concerns to broad ideas. Your comments have led us to revise text and several figures of the paper for clarity and detail, include a richer body of literature, and insert additional statistical analyses and robustness tests. While the results of the paper have not qualitatively changed, the paper is now stronger. We hope that you find this revision significantly improved as a result of the changes, and will consider recommending it for publication.

In the document below, we have presented text in grey *italics* to quote a given review or editorial comment verbatim, in its entirety, and in the order of the review. Following each quote, we address the comment by discussing how we improved our manuscript to the expected satisfaction of the Reviewers and Editor. As the Editor and Reviewers will see, we took all comments generated in the review process serious, thoroughly addressed each and every one of them, and made revisions to our manuscript as a result.

Our responses are organized into the following sections:

- Response to Reviewer 1
- Response to Reviewer 2
- Response to Reviewer 3
- Response to Reviewer 4

Sincerely,

Weihua Li, Sam Zhang, Zhiming Zheng, Skyler J. Cranmer, and Aaron Clauset

Response to Reviewer 1

The work aims at assessing the network effects of research collaborations on the productivity (measured by number of publications) and prominence (measured by high impact publications) of individual scientists. To that purpose the authors apply two network models, by which they try to explain also: i) whether differences in collaboration behaviour across gender have an effect on productivity and prominence; and ii) the extent of transferability of network capital from senior to junior scientists. The authors conclude that there is a strong relation among the variables explored.

The questions posed by the authors are intriguing and significant to the field of research assessment. Unfortunately, while their network models appear quite sophisticated, they apply them on quicksand. It is evident that the authors are not familiar with the fundamentals of the microeconomic theory of production, and the basics of scientometrics, which led them to choose an invalid indicator of productivity, and incur in several methodological flaws.

Addressed: We thank the Reviewer for their thoughtful comments and their concern, and we believe that addressing them has improved the clarity of the manuscript and its relationship with existing relevant literature. We note here that Reviewer 2, Reviewer 3, and Reviewer 4 all agreed with our basic methodological choices of scholarship evaluation in what to measure and how to measure it. In our responses below, we have endeavored to more clearly motivate and justify our choices, and we have revised the manuscript in a number of places in order to also make these rationales more clear for the reader.

I had a presentiment of that from the very beginning, when the authors stated that (page 2, line 42): “Scientists at elite institutions also receive disproportionately more funding than those at less prestigious institutions, which may enable greater scientific productivity”. As a matter of fact, productivity is a ratio of output to input. To raise productivity the marginal increase of output needs to exceed the marginal increase of input.

Addressed: Many studies have explored methods for assessing and allocating the contributions of individual researchers on collaborative publications, including assigning equal weights to all collaborators, ranking author contributions by the ordering of the author list¹, collectively allocating credits to authors based on citation patterns^{2,3}, and implementing new norms to require authors to provide contribution statements⁴. Some of the most common analytic solutions are to “normalize” publication and citation counts according to the number of authors⁴⁻⁷. However, untangling the contributions of individuals or the factors that shape

their individual contributions in team science has proven difficult, in part because these adjustments are at the level of individual papers, and hence cannot account for the characteristics of individual authors, the characteristics of their coauthors, or their coauthors' coauthors, etc. Analyses based on these approaches struggle to relate the effects of collaboration on scientific careers, which requires a person-level network, rather than a paper-level adjustment. Hence, our understanding of the collective efforts of teamwork over time, and the joint intellectual contributions to creativity⁸, remain poor.

We acknowledge the Reviewer's point about the definition of "productivity" in traditional economic systems. In our paper, our use of the term "productivity," meaning the number of papers published per faculty. This definition and our use of it follows 50 years of convention in the sociology of science and the science of science. For instance, our usage agrees with the classic papers such as Long et al. (*American Sociological Review*, 1978)⁹, and Dundar et al. (*Research in Higher Education*, 1998)¹⁰, as well as more recent studies such as Larivière et al. (*Nature* 2013)¹¹, Way et al. (*PNAS* 2019)¹², and Fortunato et al. (*Science* 2018)¹³. The word "productivity" has different meanings in different contexts, and we are explicitly using it in the traditional sociology of science sense rather than the economics sense.

To make this distinction and the motivation for our use of the term clear for the reader, in the revised manuscript, we have (i) changed our wording in the early introduction to avoid using the word "productivity" at all until it is defined, and (ii) added new discussion that explicitly notes both the definition we use, its historical grounding, and deep convention in the sociology of science literature.

The authors' definition of individual scientists' productivity, as the number of publications in a period of time, is unacceptable, as publications are not all worth the same. All others equal, two automobile manufacturers producing the same number of cars, one producing small utility cars and the other SUVs, do not have the same productivity. The authors should have used total impact in place of the number of publications.

Addressed: We again acknowledge the Reviewer's perspective on the traditional meaning of "productivity" in economic systems. In the sociology of science, productivity is not traditionally defined as a measure of impact, and so that usage would be inconsistent with the science of science literature, and, moreover, would not allow us to address our research questions. Rather, productivity in the context of the sociology of science – that is, the study of scientists and their scholarly activities and interactions – is traditionally defined as a simple count of publications, each being treated as equal to the others. Examples of this usage are numerous,

and include Crane (American Sociological Review, 1965)¹⁴, Long, (American Sociological Review, 1978)⁹, Allison et al. (American Sociological Review, 1982)¹⁵, Fox (Social Studies of Science, 1983)¹⁶, Taylor et al. (Organizational Behavior and Human Performance, 1984)¹⁷, Rodgers et al. (Journal of Applied Psychology, 1989)¹⁸, Allison et al. (American Sociological Review, 1990)¹⁹, Dundar et al. (Research in Higher Education, 1998)¹⁰, Fox (Social Studies of Science, 2005)²⁰, Van Arensbergen et al. (Scientometrics, 2012)²¹, Way et al. (PNAS, 2019)¹², and Huang et al. (PNAS, 2020)²². This extensive literature locates the emphasis on the *contributions* to science, rather than the attention that those contributions receive, which is what the term “impact” measures. The bibliometric literature has explored a variety of adjustments to “raw” productivity counts, including ones that adjust for the visibility of the publication venue (e.g., the journal impact factor) or for the number of coauthors. However, each of these makes assumptions with uncertain external validity, or confounds productivity with impact in ways that complicate interpretation. Our focus on the simple counts avoids these complications, but does introduce some limitations, which we discuss in the manuscript itself. Our definition of highly-cited papers is also consistent with past work. For instance, Uzzi et al. (Science, 2013) defined “hit” papers as those in the upper 5% of citations received across the whole dataset, as measured by total citations through 8 years²³. Ahmadpoor et al. (Science, 2017) defined “home run” papers as those being in the upper 5% of citations received in that field and year²⁴. More fundamentally, in the fields we study, simple publication counts are part of the normative and formal evaluation of scholarship, e.g., at tenure, and hence the publication counts (productivity) that we study are both a standard measure in the field and a variable with practical relevance, for how scholars in these fields themselves define and assess scholarship.

We also note that several authoritative rankings explicitly use publication counts and the number of field-normalized highly-cited works as indicators of research quality and institutional prestige, including the Nature Index and CWTS Leiden Ranking. The Nature Index website states, “*the index tracks contributions to research articles published in 82 high-quality natural science journals, chosen by an independent group of researchers*”, and “*the Nature Index provides absolute and fractional counts of article publication at the institutional and national level*”. The Nature Index does not weigh papers by their publication venues. In other words, a paper published in a prestigious general science journal like *Nature* contributes an equal weight as a paper published in field journals such as *Chemical Communications* or *eLife*. The Leiden Ranking uses the total number of publications of a university as its first indicator, and explains on the official website that “*only publications of the Web of Science document types article and review are taken into account*”. The Leiden Ranking also defines two indicators of impact $P(\text{top } 1\%)$ and $PP(\text{top } 1\%)$ to be “*the number (P) and the proportion (PP) of a university’s publications that, compared with other publications in the same field and in the same year, belong to the top 1% most frequently cited*”. These rankings embody a synthesis

of current norms around measuring and evaluating scholarship, and they prominently include both measures related to the number of publications and measures related to the number of field normalized highly-cited publications, which are the two dimensions our study focuses on.

To help make our motivation for selecting these measures more clear for the reader, we have added clarifying text to the Introduction that acknowledges the range of possible measures that can be used to quantify scholarly activities, and explicitly ground our choice in the conventions of the sociology of science literature.

Moreover, the authors apply the full counting method, instead of the fractional one. Assuming that a publication corresponds to a scientific discovery, all others equal, to each co-author one should assign a proportion of that publication, not to favour them vis-à-vis colleagues producing a publication with fewer co-authors. The number of discoveries is two, and not the sum of co-authors.

Addressed: The issue raised by the Reviewer is important, and is entirely mitigated by the fact that we only count publications for the first and last authors. This approach eliminates the variable effects of longer or shorter author lists on productivity counts, by dropping all middle coauthors. As a result, our analyses say little about middle-author collaboration network effects – a limitation that we discussed in the original version of the manuscript. Under this first/last author approach, each paper adds to the total productivity of exactly and only two authors. If we assigned an equal fraction of a publication count to each of these two authors, productivity counts would simply be divided in half, which will not qualitatively change the results.

To clarify this subtlety for the reader, we have added a statement in the Methods section and we have revisited and clarified our previous explanation of our first/last author counting approach in the Introduction and Results sections, and our discussion of this approach's limitations in the Discussion section.

Furthermore, it seems that the authors are working at top level 0 of MAG field classification. If it is true, citations need to be field-normalized at finer-grained level, because the citation behaviour varies a lot within top level 0 fields.

Addressed: A large number of studies from bibliometrics and science of science develop their main findings on pure citation counts without normalization, e.g., Wuchty et al. (Science,

2007)²⁵, Larivière et al. (PLoS One, 2015)²⁶, Sinatra et al. (Science, 2016)²⁷, and Wang et al. (Research Policy, 2017)²⁸. In addition, other commercial indices such as the impact factors of journals by Clarivate’s Web of Science and the global university ranking by the Leiden Ranking also use unnormalized citation counts. We agree with the Reviewer that normalizing citations across field and year is essential to establish a more robust measure of impact, and we have done this in our definition of highly-cited papers, where we control for field and year.

Finer-grained field levels beyond the currently used level 0 are difficult to operationalize. The MAG “fields of study” dataset assigns disciplines to papers with a hierarchical structure of levels. We can see that the number of fields explodes and does not follow a predictable pattern beyond level 2 (Fig. R1), which leads us to focus our discussion here on level 1 fields. However, even using level 1 to normalize impact introduces additional confounding factors in this context. Many papers belong to several level 1 fields, and the cohort of papers from one particular level 1 field are usually classified into several level 0 fields. Specifically, the majority of papers (over 99%) are assigned to exactly one level 0 field, while a paper on average is assigned to 1.82 level 1 fields.

Figure R1: **Number of fields in MAG levels.**

Such data structures of field annotations make normalizing impact in finer-grained fields difficult and perplexing. For instance, if a physics (level 0 field) paper also belongs to nanotechnology (level 1 field) and condensed matter physics (level 1 field), this raises the concern of whether a paper should be regarded as highly-cited if it is highly-cited in both fields or just in one of the two fields. Moreover, the hierarchy is not nested, and papers in the same level 1 field can be associated with different level 0 fields. For example, many nanotechnology

papers are classified as chemistry (level 0 field), and many condensed matter physics papers are labeled as materials science (level 0 field). When defining whether this physics paper is highly-cited, it is not clear whether it should be compared to nanotechnology papers that also fall into physics, or all nanotechnology papers regardless of their level 0 field labels.

We also tried to re-define highly-cited papers in level 1 fields. We first regard a paper as highly-cited when it receives the upper 5th percentile of citations in at least one level 1 field. Per this definition, the proportion of highly-cited papers are 5.1% for computer science, 5.3% for mathematics, and 5.5% for physics, compared to the 5% background rate. We then define a paper as highly-cited when it receives the upper 5th percentile of citations in all level 1 fields. The proportion of highly-cited papers are 3.7% for computer science, 3.6% for mathematics, 3.8% for physics, compared to the 5% background rate. Either attempt has obvious drawbacks in its definition and substantially alters the proportion of highly-cited papers from the designated background rate.

To clarify these points for the reader, we have added a brief discussion in the Methods section explaining the rationale for and trade offs implicit in the field level data in MAG, as they relate to normalizing citation counts with respect to field and year, and how these choices impact our modeling results.

The accuracy of authorship disambiguation is critical in these kinds of analysis. It is important then to provide more details on the precision and recall of the disambiguation algorithm applied.

Addressed: The Microsoft Academic Graph (MAG) team published an article in Quantitative Science Studies explaining the methodology they used in collecting and curating the MAG dataset²⁹. We quote their methodology of author name disambiguation here that “*both machine learning and crowdsourcing approaches are employed for MAG*”, and that “*MAS deliberately takes the opposite direction and decides to err on the conservative side, namely, publications bearing the same author name are not assigned to the same author node in MAG unless such assignments can exceed a 97% confidence threshold based on the machine learning algorithm. This artificially high threshold leads to author underconflation, where publications by the same author are split into multiple clusters if the variants in coauthors and topics are different enough to lower the confidence below the threshold. This design choice leads to the fact that the publication count of an author node in MAG can only be lower than the actual number of publications by the real-world author, complementing the upper bound estimates from systems that overconflate author publication records*”. (MAS stands for Microsoft Academic Services.)

As the MAG team have said, their author name disambiguation method has high accuracy when assigning papers to individual authors. Such approach facilitates the study of more productive researchers with highly accurate publication records. We randomly sample 50 authors among the productive mid-career researchers selected for the analyses. We manually check the sampled researchers and find that over 98% of articles are correctly assigned to individual researchers.

Finally, the authors' review of the literature and the methodology section lack a large number of pertinent studies published in bibliometrics journal (e.g. Scientometrics, Journal of Informetrics, Research Evaluation, etc.), which confirms the scarce familiarity of the authors with scientometrics and its consequences on the soundness of the work. Only two works out of 42 in the references list are published in bibliometric journals.

Addressed: In addition to the points about the other fields, we thank the Reviewer for the suggestion, and in the revised manuscript we have added additional references from the bibliometrics literature to help contextualize the choices we made in what to count, how to count it, and the theoretical basis for our study design, which we agree will strengthen the manuscript for the reader:

- 13. Dehdarirad, T., Villarroya, A. & Barrios, M. Research on women in science and higher education: a bibliometric analysis. *Scientometrics* 103, 795–812 (2015).
- 19. Abramo, G., D'Angelo, C. A. & Murgia, G. Gender differences in research collaboration. *Journal of Informetrics* 7, 811–822 (2013).
- 20. Bozeman, B. & Corley, E. Scientists' collaboration strategies: implications for scientific and technical human capital. *Research Policy* 33, 599–616 (2004).
- 26. Uddin, S., Hossain, L., Abbasi, A. & Rasmussen, K. Trend and efficiency analysis of coauthorship network. *Scientometrics* 90, 687–699 (2012).
- 36. Nicolaisen, J. Citation analysis. *Annual Review of Information Science and Technology* 41, 609–641 (2007).
- 41. Lozano, G. A., Lariviere, V. & Gingras, Y. The weakening relationship between the impact factor and papers' citations in the digital age. *Journal of the American Society for Information Science and Technology* 63, 2140–2145 (2012).
- 50. Van Arensbergen, P., Van der Weijden, I. & Van den Besselaar, P. Gender differences in scientific productivity: a persisting phenomenon? *Scientometrics* 93, 857–868 (2012).

Response to Reviewer 2

The research is timely and provides a novel contribution to the literature. The findings that network effects have more influence on publishing quantity and impact than other variables like gender and institutional prestige support emerging work in this area. The perspective of the literature review, and interpretation of the results, are consistent with recent research showing collaboration networks can largely explain differences in publishing rates. The current manuscript advances this argument by also looking at article impact (citation rates), beyond mere quantity of publication. There is also more refined data analysis in excluding low rate publishers and focusing on mid-career scientists and their senior collaborators. Further, the current paper controls for confounding variables.

Addressed: We appreciate the Reviewer's encouragement of our work, and hope that we have amended the manuscript to his/her full satisfaction.

The only area I can see improvement in is the details of the methodology. How were the publication years selected? How were the STEM fields selected? That is, why is engineering excluded but medicine included?

Addressed: We select the publication years based on the following criteria. First, the version of the Microsoft Academic Graph (MAG) data we used collects papers published from the early 1900s to 2019. To define highly-cited papers, we use the number of citations accrued two years after publication, which means that we can only annotate highly-cited works that are published up to 2017. Because the mid-career researchers used in our study have 15 years of publishing career, which means they should have their first papers published before 2003. For each year, we compute latent variable models using all prior publications, and sample papers using a bootstrap sampling procedure for each estimation. Because we have to eliminate any tree-like structures in the coauthorship network, we need a decent number of eligible researchers within a given field. Thus the data points are too scarce for the network models in very early years, especially for relatively small fields. For some of the selected fields, such as mathematics, we cannot operationalize the estimation of models in 1970 due to the above reasons, but we can do that since 1975. Given these considerations and restrictions, we select mid-career researchers that published their first papers from 1975 to 2003.

The historical productivity of a specific field affects the time period eligible for the network models, as we have discussed. Thus, our approach will work better for larger disciplines than smaller ones. Medicine and biology are the largest fields from which we can select a rich cohort of productive mid-career researchers to analyze. Engineering consists of articles from

many divergent research areas, and instead we used computer science, which converge to more focused research themes to represent engineering-related fields. We show the number of papers for all STEM fields from the Microsoft Academic Graph dataset after our data cleaning procedures, and we can see that the selected fields encompass the majority of papers (Fig. R2).

Figure R2: **Number of papers in each STEM field in the Microsoft Academic Graph dataset.** Red bars represent the number of papers published in fields that are selected for our study.

To clarify these points for the reader, we have added a brief discussion in the Methods section that *“Among them, we select six scientific fields representative of the traditional science, technology, engineering and mathematics (STEM) domains: biology, chemistry, computer science, mathematics, medicine, and physics. These fields publish the majority of research papers in science and technology domains (see Supplementary Fig. 1).”*

Response to Reviewer 3

The manuscript titled “Untangling the network effects of productivity and prominence among scientists” by Li et al. is a well written and well-documented analysis integrating collaboration networks, institutional prestige and gender information to explain the variation in observed productivity and prominence across a large set of STEM researcher profiles based upon the Microsoft Academic Graph dataset.

A key result based upon their analysis that controls for network effects, is that productivity is low-variance and that prominence is high-variance; and also the stability of the estimated productivity and prominence parameters over the career; and the statistically indistinguishable parameters by gender. In general, the manuscript exhibits robust and insightful results, a strong methodology, and strong visualization. Yet there are a few concerns regarding the true generalizability of the results as stated.

Addressed: We thank the Reviewer for complimenting the results, methodology and visualization of our work, and hope that we have amended the manuscript to the Reviewer’s full satisfaction.

First, the choice of constructing the social network based upon first and last authors of papers requires additional justification and robustness check. One reason for this is that it means that many mid-career scientists who are commonly found in the middle positions are omitted, and so it’s not clear to what extent this affects the results of the study which the authors state is aimed at identifying differences in productivity an prominence of mid-career researchers; see Zeng XH, Duch J, Sales-Pardo M, Moreira JA, Radicchi F, Ribeiro HV, Woodruff TK, Amaral LA. Differences in collaboration patterns across discipline, career stage, and gender. PLoS biology. 2016 Nov 4;14(11):e1002573. An additional pertinent analysis relating team size and annual productivity across the career are reported in Petersen AM, Riccaboni M, Stanley HE, Pammolli F. Persistence and uncertainty in the academic career. Proceedings of the National Academy of Sciences. 2012 Apr 3;109(14):5213-8. A similar confounder occurs at the data selection stage, where it is indicated that roughly only 1 in 2 MAG articles contain affiliation data, and so the authors exclude those without affiliation data. This seems like a large sample selection bias combined with the first-last author criterion, that highly conditions the results.

Addressed: The Reviewer has proposed very good suggestions here, and we agree with the Reviewer that having dropped middle authors in the theoretical models would limit the scope of our analyses. However, as what we have argued in the manuscript, the scientific norms around

middle authorship are complex and variable, and they are also confounded with team size. Therefore, including them on equal basis with other publications would lessen the generality of our results. In contrast, dropping middle authors does not mean that we are omitting a substantial proportion of collaboration network, but it allows us to extract a more definitive signal related to advisor-advisee relationships, which are casually more likely to relate to notions of social capital and inter-generational transmission. In addition, many recent studies have explicitly focused on first/last authors, e.g., Nielsen, et al., (Nature Human Behaviour, 2017)³⁰, Sauermann, et al., (Science Advances, 2017)⁴, Fox, et al., (Ecology and Evolution, 2018)³¹, Ni, et al., (Science Advances, 2021)³², and Jiménez-García, et al., (American Journal of Ophthalmology, 2022)³³.

In order to make these tradeoffs more clear for the reader, we have added additional discussion of this design choice in Introduction “*A number of recent studies have shown that inequality in social networks and collaborations may relate to gender disparity and affect career outcomes for women^{34–39}, particularly in science, technology, engineering, and mathematics (STEM) fields^{40–43}.*” “*Via collaboration, networks correlate with the unequal provision of “scientific and technological human capital” across researchers⁴¹, shape the academic career of researchers⁴⁴, and can conceal underlying inequalities in formal evaluations like tenure⁴⁵.*”, and Methods “*There are circumstances where this norm does not apply, e.g., in specific subfields where authors are listed in the alphabetic order, or when there are multiple “first” or “last” authors due to equal contribution flags, as well as in some large collaborations. Our first-last author counting scheme eliminates the effects of large author lists and the relevance of fractional counting, at the expense of potentially under-counting contributions and effects of middle-authorship.*”

Second, the language regarding “network effects” is a bit of an embellishment (“coauthorship effects” may be more suitable here), as the authors are primarily capturing the first-order network (ego-network) of researchers, as opposed to the entire collaboration network which the authors convey in the abstract and introduction.

Addressed: We have extensively used the term “coauthorship networks” in our manuscript, e.g., in the abstract we said “*We find that gendered differences in the productivity and prominence of mid-career researchers can be largely explained by differences in their coauthorship networks.*”, in order to emphasize the fact that a researcher may have many different collaborators which forms a complex web of coauthorship ties. Some studies of coauthorships, e.g., Li, et al., (Nature Communications, 2019)⁴⁶, use a dichotomous variable to indicate whether a researcher experienced a certain collaboration pattern rather than focusing on the entire

network of researchers. Although we didn't consider the full coauthorship network, as the Reviewer correctly points out, our proposed method is still inherently a network model, because an individual researcher's parameter estimates depend on those of his/her collaborators, and his/her collaborators' collaborators, etc. Moreover, coauthorship between middle authors in a large team does not necessarily involve personal relations, whereas first-last authors usually have deep interactions and knowledge transfer.

Moreover, the effects that we are estimating are really network effects because the ego-networks are overlapping, and hence our estimates of one author's latent parameters depend on the estimates of their coauthors' latent parameters, which further depend on the their coauthors' parameters, etc. As we have explained in the paper, this allows our model to control for the network effects of one's collaborators' productivity or prominence on his/her own productivity or prominence. As such, we think using the term "network effects" may better reflect the networked nature of our latent models.

Another less central issue is the language that suggests that their method is correct and that traditional measures are incorrect: "efficacy of these two measures by characterizing their correlation with other "uncorrected" measures and time-related dynamics for individual researchers", which is a bit presumptive. Possibly this is just an issue of word choice, and so adjusted could replace corrected and ameliorate this issue. Moreover, in order to assess the efficacy in the case of theta, a better baseline for comparison would be the solo-authored papers by the same author (for which $\theta_j = 0$ by construction).

Addressed: The Reviewer makes a very good suggestion here. When saying "uncorrected", we actually refer to crude publication and citation counts, to distinguish from our proposed latent productivity and prominence variables, which, however, as the Reviewer said, may inadvertently implied that other measures are "not correct". We have therefore replaced "uncorrected" by "unadjusted" in our revision, and reviewed the entire manuscript to ensure that our choice of words does not imply that we are "right" and all others are "wrong".

We also use the solo-authored papers among mid-career researchers to assess the efficacy of θ . Among these mid-career researchers, 90,583 of them published at least one solo-authored paper. There are 48,484 researchers with negligible estimated values of $\theta < 10^{-3}$, and 84.8% (41,102) of them published no highly-cited solo-authored paper up to the mid-career. To test if the prominence model (θ) better predicts researchers' impact, we use a random null model to estimate a baseline of publishing highly-cited work. The null model randomly reshuffles solo-authored papers among authors, and we find that 79.0% of researchers with $\theta < 10^{-3}$ published no highly-cited solo-authored papers, 5.8% lower than the empirical percentage.

This suggests that low estimated θ value moderately increases the precision in predicting whether researchers have highly-cited solo-authored papers or not. We have incorporated this in the revision for the reader and thank the Reviewer for pushing us in this direction.

One final issue are the results comparing early-career researchers collaborating with senior researchers at elite institutions. Clearly there is a section issue that explains the result, that being that high lambda or that researchers are more likely to be at elite institutions.

Addressed: The Reviewer makes a good point here. In the manuscript, we added “*This may be largely due to a selection effect that high λ or θ senior researchers are more likely to work at elite institutions, reflecting inequalities of having access to important social networks among early-career researchers.*” We wanted to highlight how environmental prestige affects the formation of junior-senior author collaborations, which adds up to inequalities in scientific careers.

In summary, despite the above concerns, this work merits strong consideration for publication as certain findings (eg regarding gender) are in very near alignment with previously reported findings, and so given that appropriate comparative approaches are used in this analysis, there is much reason to trust the differences to be robust to the underlying caveats.

Addressed: We thank the Reviewer for finding our methodology and results robust and for recommending publication of our manuscript. We hope that we have been able to revise our paper to his/her full satisfaction.

Additional Comments:

- *Figure 1: Panel B - A scatter plot is not the most informative plot as there are too many datapoint to appreciate the density, and so a 2-d density plot would be better suited. Panel C - diagonal elements should be removed as they render other relevant relationships difficult to distinguish in magnitude.*

Addressed: The Reviewer made a very good suggestion of improvement regarding Figure 1 in the main text, and we present the revised Figure 1b in Fig. R3 and Figure 1c in Fig. R4. As the Reviewer can see, we add on top of the original scatter plot of Figure 1b with the inset density plot. We retained the scatter plot because the R package ggExtra depends on the scatter plot to insert the yellow marginal bars, and that most mid-career researchers concentrate in a small area of the parameter space ($\lambda < 0.8$ & $\theta < 0.2$). For Figure 1c, we removed the colors and correlation values on the diagonal as suggested by the Reviewer. We believe that these modifications make the figure more informative, and we hope that the Reviewer would agree.

Figure R3: **Panel b of Figure 1 in the main text.** We added the contour density plot in the inset.

-
- *Statements including “papers per year” should be rephrased as “first/last-authored papers per year”.*

Figure R4: **Panel c of Figure 1 in the main text.** We removed the diagonal in the previous version of the manuscript.

Addressed: We think the Reviewer’s suggestion could make the definition more accurate, and we have rephrased “papers per year” as first/last-authored papers per year” as advised by the Reviewer.

-
- *I found the statement “indicating that they are performing as desired in controlling for the network effects of collaboration” on line 133 odd yet perplexing, and so I believe it merits elaboration & clarification.*

Addressed: We think that after controlling network effects, eligible parameters to model productivity and prominence of individual researchers should be largely independent of career-wise productivity. The low correlation between the estimated parameters and total productivity proves our hypothesis, and may be better indicators to assess a researcher’s performance.

To better clarify the idea, we added in the main text of the manuscript: *“this implies that after controlling for the network effects of collaboration, the latent parameters could indicate the productivity and prominence of individual researchers per unit time period.”*

-
- *There are a number of arbitrary thresholds used in the manuscript (eg “at least 3 papers by the year of relevant collaboration” and “the early-career researcher is 5 or fewer years since their first publication, and the senior coauthor is 6 or more years since their first publication.”) which further raise the issue of robustness: how were these thresholds chosen and to what degree to results change if they are for example doubled?*

Addressed: Our work starts with theoretical models, but the proposed parameters need to be estimated from empirical data, which, as the Reviewer can see, are usually complex and noisy. The estimated latent variables for researchers with extremely low productivity, e.g., those that have published only one paper by the time of relevant year, are not really informative and cannot predict their future performance (see Fig. 3). Also, these researchers have not really established a mature and well-connected social network, on which our models rely. On the other hand, if we only retain highly productive researchers, the resulting sample size might be too limited for robust statistical analyses.

We are also faced with many decisions with empirical data, such as defining junior or senior researchers. Given our experience in STEM fields, for a student on a typical doctoral program that lasts five years, he/she may very likely to publish his/her first paper in year 3-5. Junior researchers may also take on a post-doc position for another 2-3 years after graduation. Thus, we believe that a large number of researchers may have obtained faculty positions 6 years after their first publications.

To ensure that our results are not sensitive to our choice of thresholds, we conduct several robustness checks as suggested by the Reviewer. We repeat the analyses by selecting senior collaborators that have at least 6 publications and at least 10 publishing career years by the time of relevant collaboration. We think that the results hold in the new threshold settings (Fig. R5).

To clarify this for the reader, we insert a brief comment in the revised manuscript: *“We also test the robustness of our results by selecting senior collaborators with at least 6 publications and at least 10 publishing career years by the time of relevant collaboration, (Supplementary Fig. 15), and we find that the different thresholds do not qualitatively change our findings.”*

Figure R5: **Replicating results for Fig. 3 in the main text regarding collaborations.** We use new thresholds to identify collaboration and aging effects. The selected senior collaborators are defined as those having at least 6 publications and at least 10 publishing career years by the time of relevant collaboration.

Response to Reviewer 4

This paper consists of an innovative analysis of STEM researchers' collaboration networks and how they relate to differences in productivity and prestige depending upon gender and institutional status.

My comments below mainly focus on the extent of the contribution and the analytic approach.

1.) Contribution

The authors have discovered that gender gaps in productivity and prominence are mitigated when accounting for collaboration networks (operationalized in terms of first-author and last-author pairs). The dataset constructed and the analytic approach are quite innovative. The authors also have discovered that scientists working at elite institutions have a distinct advantage, and that this disparity is not mitigated when controlling for network effects. In fact, researchers working in a non-elite institution appear to be at a distinct disadvantage even when working with a prominent co-author.

Addressed: We thank the Reviewer for finding our constructed data and analytic approach innovative. We hope that we have amended the manuscript to the full satisfaction of the Reviewer.

Plenty of extant research has shown that women have less access to important social networks compared to men, and that these gender gaps in network access and brokerage help to explain gender differences in career outcomes (Belliveau, 2005; Greguletz et al., 2019; Ibarra, 1992, 1997; for a meta-analysis, see Fang et al., 2020) and in STEM in particular (Abramo et al., 2013; Bozeman & Curley, 2004; Collins & Steffem 2019; see Casad et al., 2021, section entitled "Social Capital" for a short review of gender, social capital, and STEM). Thus, although the findings reported here are compelling, it would be helpful to understand further how these findings contribute above and beyond other past work on gender and social networks and the concomitant effects on careers. In fact, I think that the dataset in this paper provides a rich portrait of collaboration networks and makes some unique contributions, especially in the STEM context. Overall, I recommend that the authors elaborate on how these findings make a unique contribution in terms of understanding the relationships among gender, social capital/networks and career outcomes in STEM.

Addressed: We thank the Reviewer for providing a rich collection of references and we have incorporated them into the introduction of the manuscript: "A number of recent studies have shown that inequality in social networks and collaborations may relate to gender disparity and

affect career outcomes for women^{34–39}, particularly in science, technology, engineering, and mathematics (STEM) fields^{40–43}.” We think these narratives improve the literature review of the paper, and emphasize the important role social networks play in shaping inequalities for women in science.

As we have discussed in the manuscript, *“In particular, collaboration network effects can explain both the persistent gendered inequalities among mid-career researchers in productivity and prominence, and a considerable portion of the observed inequalities between researchers working in more or less elite environments.”* The unique contribution we have made on top of the literature of gender disparity is that mid-career women’s latent productivity λ and prominence θ are essentially comparable to men. The gendered gap in crude publication and citation counts are mostly explained by the effects of social networks. Being able to untangle the network effects provides fresh perspectives in understanding gender inequalities.

To better connect our findings about gender with the rich literature the Reviewer has recommended, we have inserted a brief discussion in the revised manuscript: *“By providing new individual parameters after adjusting network effects, our findings highlight the importance of social networks in shaping scholarly gender differences among mid-career researchers, which contributes to the abundant literature on potential causes and effects of gender disparity in science, including academic culture⁴⁰ and homophily^{36,37}.”*

Furthermore, the authors analyze gender gaps and institutional (elite/non-elite) gaps separately. The gaps are, in fact, much larger for institutional status versus gender. Did the authors test an interaction between gender by institutional status and parse those findings? I would be curious to see whether institutional status trumps gender. Based on the main effects of gender versus institutional status, it appears that one’s institution has a much stronger effect on career outcomes compared to gender.

Addressed: We think the Reviewer makes a really good suggestion here. To examine this possibility, we test the interaction between gender and institutional prestige among mid-career researchers. In Fig. R6, we find that the prestige of institutions has stronger effect on researchers’ productivity and prominence, for both crude measures and latent parameters. In particular, the gender and institution effects are negligible for latent productivity λ (Fig. R6b), while institutional prestige has stronger effect on latent prominence θ compared to gender. These findings very well prove the Reviewer’s hypothesis.

In the main text of the manuscript, we add *“In addition, we test the interaction effects of gender and institutional prestige on the performance of mid-career researchers. We find that the prestige of institutions has relatively stronger effect on researchers’ productivity and*

Figure R6: **Interaction of gender and institution effects for mid-career researchers.** We test the **a** publications, **b** latent productivity λ , **c** citations, and **d** latent prominence θ for mid-career researchers by the interaction of institution and gender.

prominence than gender, for both crude measures and latent parameters (see Supplementary Fig. 12). In particular, both gender and institutional prestige have negligible effects on latent productivity λ , while institutions appear to have stronger influence than gender on latent prominence θ . These findings are in line with recent studies that suggest the disproportionate productivity of elite researchers are largely due to their labor advantage in which environmental prestige has a rather limited role⁴⁷.

2.) Analytic approach

Overall, I applaud the authors for conducting such a thorough and comprehensive analysis. Their efforts are impressive. I raise the following issues/questions below to help readers gain clarity over their sampling and analytic decisions made.

Addressed: We thank the Reviewer for such encouraging comments of our research, and hope that we have fully addressed all of the points raised by the Reviewer in the revised manuscript.

-Thresholds:

The choices for certain thresholds used in the analyses were a bit unclear. For example, why is prominence modeled as the top 8% of citations? Why not 5% or 10%, for instance? Similarly, for the elite institution threshold, why top 10 as opposed to top 5, top 15 etc.? Similarly, why did the authors choose 10 publications within 15 years for their sample? This seems like a fairly low threshold for productivity.

Addressed: The Reviewer raises a good set of questions here. As the Reviewer has seen, we are faced with many decisions in the definition of variables using empirical data, including highly-cited paper, elite institution, productive mid-career researchers, and more. For instance, the percentage of highly-cited papers should be constrained within a realm so that it is small enough to identify truly impactful work while large enough to give non-zero estimated latent prominence to an appropriate cohort of researchers. Also, the selected mid-career researchers should have enough papers to obtain relatively reliable latent parameters, but not overwhelmingly productive so that we could retain a large enough sample size. These are basically the considerations when we dial the thresholds in empirical data.

We test the robustness of our findings using a range of new thresholds. We recompute the prominence model using 5% and 10% thresholds as highly-cited papers in computer science, mathematics, and physics (Fig. R7). We find that the statistical distribution of estimated parameters does not change qualitatively as we change thresholds, and θ still follows a heavy-tail distribution and has little correlation with λ .

We then test the robustness of our results by defining the top 20 institutions in a given field as elite institutions (Fig. R8). The results hold qualitatively, in that the productivity of mid-career researchers from elite institutions is moderately higher than those from non-elite institutions, whereas the prominence of researchers from elite institutions substantively outperform those from non-elite institutions. The disparity in productivity can be mitigated after controlling for network effects, but the gap in prominence can only be partially explained by network effects.

Finally, we replicate our analyses on gender disparity by selecting mid-career researchers with at least 20 publications. The findings hold with the original threshold of 10 publications. After controlling for network effects, the gendered gap in productivity and prominence can be largely explained. These robustness tests further validates our selection of thresholds, and that the findings are not sensitive to parameter settings.

To clarify these points for the reader, we have added brief discussions of the robustness tests in the revised manuscript: *“The distribution of θ does not qualitatively change*

Figure R7: **Replicating latent prominence model.** We re-compute the θ values of mid-career researchers for **a** computer science, **b** mathematics, and **c** physics by defining highly-cited papers as those receiving the the **a-c** upper 5th and **d-f** 10th percentile citations two years after publication.

when we alter the threshold of highly-cited papers (Supplementary Fig. 6).” “We also test the robustness of our findings by selecting mid-career researchers with at least 20 publications (Supplementary Fig. 13) and repeating the analysis by randomly sampling a tertile of researchers (Supplementary Fig. 14), showing that these different choices do not change the qualitative nature of our conclusions.” “In addition, we find that the results do not qualitatively change when we modify the number of selected elite institutions to the top 20 (Supplementary Fig. 16).”

-Sensitivity analysis:

Related to the point above about sampling, it would be interesting to see whether the patterns found hold when examining the sample by quartiles or tertiles given the large differences in productivity among the researchers sampled.

Addressed: The Reviewer makes a good suggestion here. We randomly sampled a tertile of

Figure R8: **Replicating results for Fig. 4 in the main text regarding institutions.** In this analysis, we select the top 20 research institutions as elite institutions, instead of top 10 institutions used in the main text.

mid-career researchers and replicate the analyses for gender disparity (Fig. R10). We find that the gendered gap in productivity and prominence can be largely explained after controlling for network effects, which validates the robustness of results reported in the main text.

We highlight these additional tests in the revised manuscript: *“We also test the robustness of our findings by selecting mid-career researchers with at least 20 publications (Supplementary Fig. 13) and repeating the analysis by randomly sampling a tertile of researchers (Supplementary Fig. 14), showing that these different choices do not change the qualitative nature of our conclusions.”*

- Effect sizes

Please report effect sizes for significant results. Given the size of the dataset, the authors report highly significant effects, but some of the absolute values of the numbers are fairly small (e.g, the means by gender of number of published papers reported on the bottom of p. 7).

Addressed: The Reviewer provides a really good suggestion here. As he/she can see, in the main text we have reported additional statistics including the t-statistic and Cohen’s

Figure R9: **Replicating results for Fig. 4 in the main text regarding gender.** In this analysis, we select mid-career researchers that have at least 20 publications, instead of 10 publications in the main text.

d-statistic, where we compare productivity and prominence across gender, coauthors, and institutional factors. We hope that these efforts could improve the quality of the report, and hope that the Reviewer would agree.

Additional comments:

- *Did the authors investigate whether the gendered make-up of the co-author pairs affected the results? I could not find a discussion of that in the paper.*

Addressed: The Reviewer made a good suggestion here. We have considered the possibility of conducting similar analyses based on the gendered pairing of collaborators, but we chose not to do it due to the sensitivity of this issue. A previous article compared the gendered coauthor pairing between junior and senior researchers⁴⁸, and reported that “increasing the proportion of female mentors is associated not only with a reduction in post-mentorship impact of female protégés, but also a reduction in the gain of female mentors”, which incurred wide criticism on social media. Given the sensitivity of the issue, while we find the topic interesting

Figure R10: **Replicating results for Fig. 2 in the main text regarding gender by sampling a tertile of researchers.** In this analysis, we randomly select a tertile of mid-career researchers.

and would have broad implications for research policy, we think it deserves a fully new and carefully designed follow-up study.

-
- *Please elaborate on the connection to parenthood status made at the top of p. 14. I don't follow how these results are related to work on parenthood status. Are the authors implying that women have fewer collaborators, and thus don't reap the benefits of collaboration, because they have more caregiving demands? That's interesting, but please elaborate on this point further, as well as how the variables collected relate to that point.*

Addressed: We were trying to explain that while we find network effects mitigates the gendered gap of productivity and prominence among mid-career researchers, our analyses do not establish any casual inferences. Other factors like parenthood, as the Reviewer correctly

points out, may disproportionately sidetrack the attention of academic women to caregiving duties and having less time to build their collaboration networks.

To clarify this point for the reader, we modified the language to “*We note that this analysis does not establish a causal relationship, and hence known causal factors, such as the gendered impact of parenthood on researchers that leads to productivity penalty for mothers as they undertake more childcare duties⁴⁹, likely influence both productivity and collaboration networks.*” Factors like parenthood might be the inherent cause of gendered gap in social networks, which, however, are beyond the scope of this study and the coverage of the Microsoft Academic Graph data.

References

Abramo, G., D’Angelo, C. A., & Murgia, G. (2013). *Gender differences in research collaboration. Journal of Informetrics, 7(4), 811-822.*

Belliveau, M. A. (2005). *Blind ambition? The effects of social networks and institutional sex composition on the job search outcomes of elite coeducational and women’s college graduates. Organization Science, 16(2), 134-150.*

Bozeman, B., & Corley, E. (2004). *Scientists’ collaboration strategies: Implications for scientific and technical human capital. Research Policy, 33(4), 599–616. <https://doi.org/10.1016/j.respol.2004.01.008>.*

Collins, R., & Steffen, N. (2019). *Hidden patterns: Using social network analysis to track career trajectories of women STEM faculty. Equality, Diversity and Inclusion, 38(2), 265–282. <https://doi.org/10.1108/EDI-09-2017-0183>.*

Casad, B. J., Franks, J. E., Garasky, C. E., Kittleman, M. M., Roesler, A. C., Hall, D. Y., & Petzel, Z. W. (2021). *Gender inequality in academia: Problems and solutions for women faculty in STEM. Journal of Neuroscience Research, 99(1), 13-23.*

Fang, R., Zhang, Z., & Shaw, J. D. (2020). *Gender and social network brokerage: A meta-analysis and field investigation. Journal of Applied Psychology, 106(11), 1630-1654.*

Greguletz, E., Diehl, M. R., & Kreutzer, K. (2019). *Why women build less effective networks than men: The role of structural exclusion and personal hesitation. Human Relations, 72(7), 1234-1261.*

Ibarra, H. (1992). *Homophily and differential returns: Sex differences in network structure and access in an advertising firm*. *Administrative Science Quarterly*, 37(3), 422-447.

Ibarra, H. (1997). *Paving an alternative route: Gender differences in managerial networks*. *Social Psychology Quarterly*, 60(1), 91-102.

References

1. Sekercioglu, C. H. Quantifying coauthor contributions. *Science* **322**, 371 (2008).
2. Hirsch, J. E. An index to quantify an individual's scientific research output that takes into account the effect of multiple coauthorship. *Scientometrics* **85**, 741–754 (2010).
3. Shen, H.-W. & Barabási, A.-L. Collective credit allocation in science. *Proceedings of the National Academy of Sciences* **111**, 12325–12330 (2014).
4. Sauermann, H. & Haeussler, C. Authorship and contribution disclosures. *Science Advances* **3**, e1700404 (2017).
5. Kennedy, D. Multiple authors, multiple problems. *Science* **301**, 733–734 (2003).
6. Allen, L., Scott, J., Brand, A., Hlava, M. & Altman, M. Publishing: Credit where credit is due. *Nature* **508**, 312 (2014).
7. Ahmadpoor, M. & Jones, B. F. Decoding team and individual impact in science and invention. *Proceedings of the National Academy of Sciences* **116**, 13885–13890 (2019).
8. Azoulay, P., Graff Zivin, J. S. & Wang, J. Superstar extinction. *The Quarterly Journal of Economics* **125**, 549–589 (2010).
9. Long, J. S. Productivity and academic position in the scientific career. *American Sociological Review* 889–908 (1978).
10. Dundar, H. & Lewis, D. R. Determinants of research productivity in higher education. *Research in Higher Education* **39**, 607–631 (1998).
11. Larivière, V., Ni, C., Gingras, Y., Cronin, B. & Sugimoto, C. R. Bibliometrics: Global gender disparities in science. *Nature* **504**, 211–213 (2013).
12. Way, S. F., Morgan, A. C., Larremore, D. B. & Clauset, A. Productivity, prominence, and the effects of academic environment. *Proceedings of the National Academy of Sciences* **116**, 10729–10733 (2019).
13. Fortunato, S., Bergstrom, C. T., Börner, K., Evans, J. A., Helbing, D., Milojević, S., Petersen, A. M., Radicchi, F., Sinatra, R., Uzzi, B. *et al.* Science of science. *Science* **359** (2018).
14. Crane, D. Scientists at major and minor universities: A study of productivity and recognition. *American Sociological Review* 699–714 (1965).

15. Allison, P. D., Long, J. S. & Krauze, T. K. Cumulative advantage and inequality in science. *American Sociological Review* 615–625 (1982).
16. Fox, M. F. Publication productivity among scientists: A critical review. *Social Studies of Science* **13**, 285–305 (1983).
17. Taylor, M. S., Locke, E. A., Lee, C. & Gist, M. E. Type a behavior and faculty research productivity: What are the mechanisms? *Organizational Behavior and Human Performance* **34**, 402–418 (1984).
18. Rodgers, R. C. & Maranto, C. L. Causal models of publishing productivity in psychology. *Journal of Applied Psychology* **74**, 636 (1989).
19. Allison, P. D. & Long, J. S. Departmental effects on scientific productivity. *American Sociological Review* 469–478 (1990).
20. Fox, M. F. Gender, family characteristics, and publication productivity among scientists. *Social Studies of Science* **35**, 131–150 (2005).
21. Van Arensbergen, P., Van der Weijden, I. & Van den Besselaar, P. Gender differences in scientific productivity: a persisting phenomenon? *Scientometrics* **93**, 857–868 (2012).
22. Huang, J., Gates, A. J., Sinatra, R. & Barabási, A.-L. Historical comparison of gender inequality in scientific careers across countries and disciplines. *Proceedings of the National Academy of Sciences* **117**, 4609–4616 (2020).
23. Uzzi, B., Mukherjee, S., Stringer, M. & Jones, B. Atypical combinations and scientific impact. *Science* **342**, 468–472 (2013).
24. Ahmadpoor, M. & Jones, B. F. The dual frontier: Patented inventions and prior scientific advance. *Science* **357**, 583–587 (2017).
25. Wuchty, S., Jones, B. F. & Uzzi, B. The increasing dominance of teams in production of knowledge. *Science* **316**, 1036–1039 (2007).
26. Larivière, V., Haustein, S. & Börner, K. Long-distance interdisciplinarity leads to higher scientific impact. *Plos one* **10**, e0122565 (2015).
27. Sinatra, R., Wang, D., Deville, P., Song, C. & Barabási, A.-L. Quantifying the evolution of individual scientific impact. *Science* **354** (2016).
28. Wang, J., Veugelers, R. & Stephan, P. Bias against novelty in science: A cautionary tale for users of bibliometric indicators. *Research Policy* **46**, 1416–1436 (2017).

29. Wang, K., Shen, Z., Huang, C., Wu, C.-H., Dong, Y. & Kanakia, A. Microsoft academic graph: When experts are not enough. *Quantitative Science Studies* **1**, 396–413 (2020).
30. Nielsen, M. W., Andersen, J. P., Schiebinger, L. & Schneider, J. W. One and a half million medical papers reveal a link between author gender and attention to gender and sex analysis. *Nature Human Behaviour* **1**, 791–796 (2017).
31. Fox, C. W., Ritchey, J. P. & Paine, C. T. Patterns of authorship in ecology and evolution: First, last, and corresponding authorship vary with gender and geography. *Ecology and Evolution* **8**, 11492–11507 (2018).
32. Ni, C., Smith, E., Yuan, H., Larivière, V. & Sugimoto, C. R. The gendered nature of authorship. *Science Advances* **7**, eabe4639 (2021).
33. Jiménez-García, M., Buruklar, H., Consejo, A., Dragnea, D. C., Fambuena, I., Hershko, S., Issarti, I., Kreps, E. O., Van Acker, S. I., Dhuhghaill, S. N. *et al.* Influence of author's gender on the peer-review process in vision science. *American Journal of Ophthalmology* (2022).
34. Zeng, X. H. T., Duch, J., Sales-Pardo, M., Moreira, J. A., Radicchi, F., Ribeiro, H. V., Woodruff, T. K. & Amaral, L. A. N. Differences in collaboration patterns across discipline, career stage, and gender. *PLoS Biology* **14**, e1002573 (2016).
35. Belliveau, M. A. Blind ambition? the effects of social networks and institutional sex composition on the job search outcomes of elite coeducational and women's college graduates. *Organization Science* **16**, 134–150 (2005).
36. Greguletz, E., Diehl, M.-R. & Kreutzer, K. Why women build less effective networks than men: The role of structural exclusion and personal hesitation. *Human Relations* **72**, 1234–1261 (2019).
37. Ibarra, H. Homophily and differential returns: Sex differences in network structure and access in an advertising firm. *Administrative Science Quarterly* 422–447 (1992).
38. Ibarra, H. Paving an alternative route: Gender differences in managerial networks. *Social Psychology Quarterly* 91–102 (1997).
39. Fang, R., Zhang, Z. & Shaw, J. D. Gender and social network brokerage: A meta-analysis and field investigation. *Journal of Applied Psychology* **106**, 1630 (2021).
40. Abramo, G., D'Angelo, C. A. & Murgia, G. Gender differences in research collaboration. *Journal of Informetrics* **7**, 811–822 (2013).

41. Bozeman, B. & Corley, E. Scientists' collaboration strategies: implications for scientific and technical human capital. *Research Policy* **33**, 599–616 (2004).
42. Collins, R. & Steffen-Fluhr, N. Hidden patterns: Using social network analysis to track career trajectories of women stem faculty. *Equality, Diversity and Inclusion: An International Journal* (2019).
43. Casad, B. J., Franks, J. E., Garasky, C. E., Kittleman, M. M., Roesler, A. C., Hall, D. Y. & Petzel, Z. W. Gender inequality in academia: Problems and solutions for women faculty in stem. *Journal of Neuroscience Research* **99**, 13–23 (2021).
44. Petersen, A. M., Riccaboni, M., Stanley, H. E. & Pammolli, F. Persistence and uncertainty in the academic career. *Proceedings of the National Academy of Sciences* **109**, 5213–5218 (2012).
45. Moher, D., Naudet, F., Cristea, I. A., Miedema, F., Ioannidis, J. P. & Goodman, S. N. Assessing scientists for hiring, promotion, and tenure. *PLoS Biology* **16**, e2004089 (2018).
46. Li, W., Aste, T., Caccioli, F. & Livan, G. Early coauthorship with top scientists predicts success in academic careers. *Nature Communications* **10**, 1–9 (2019).
47. Zhang, S., Wapman, K. H., Larremore, D. B. & Clauset, A. Labor advantages drive the greater productivity of faculty at elite universities. *arXiv preprint arXiv:2204.05989* (2022).
48. AlShebli, B., Makovi, K. & Rahwan, T. The association between early career informal mentorship in academic collaborations and junior author performance. *Nature Communications (retracted)* **11**, 5855 (2020).
49. Morgan, A. C., Way, S. F., Hoefler, M. J., Larremore, D. B., Galesic, M. & Clauset, A. The unequal impact of parenthood in academia. *Science Advances* **7**, eabd1996 (2021).

Reviewers' Comments:

Reviewer #1:

Remarks to the Author:

I recognize the enormous effort made by the authors to defend their position and to get around the real essence of productivity definition, without pondering on the implications. "Not everything that counts can be counted, and not everything that can be counted counts." The authors have chosen to count what does not count. It is true, unfortunately, that many bibliometricians perpetuate the cozy definition of productivity as the number of publications by an individual, which makes its measurement easy "to count". The question is: "does it count?". Would you promote or allocate more research funds to those who publish more or to those who have higher impact on the advancement of science, all inputs being equal? It is not simply a matter of definitions, it is the meaningfulness of definitions that matter. The authors could simply change the wording substituting productivity with intensity of publication, to be conceptually correct, but the essence would be the same. What should we do then with the findings of the effects of networking on the intensity of publication alone? I could go on with the invalid full counting of publications, and their gross field classification, which add to the original sin. Probably others can accept the authors' approach, but I cannot: science has to move forward, it cannot stay still (and above all wrong).

Reviewer #2:

Remarks to the Author:

The research is timely and provides a novel contribution to the literature. The findings that network effects have more influence on publishing quantity and impact than other variables like gender and institutional prestige support emerging work in this area. The perspective of the literature review, and interpretation of the results, are consistent with recent research showing collaboration networks can largely explain differences in publishing rates. The current manuscript advances this argument by also looking at article impact (citation rates), beyond the mere quantity of publication. There is also more refined data analysis in excluding low-rate publishers and focusing on mid-career scientists and their senior collaborators. Further, the current paper controls for confounding variables.

I believe the authors have adequately addressed concerns I raised in the first review.

Reviewer #3:

Remarks to the Author:

The authors have exhaustively addressed my initial comments, which were largely of clarifying nature.

To reiterate my initial assessment, this work is an exhaustive analysis on careers in science that integrates various analysis, modeling and visualization methods in exemplary form, and as such, merits publication.

Reviewer #4:

Remarks to the Author:

Overall, the authors have responded thoroughly to my comments about the previous version of the manuscript. I appreciate their comprehensive responses and additional analyses/results reported.

The results concerning the interaction effects of gender by institutional prestige were quite interesting. I didn't follow what the authors meant by the last sentence in the new paragraph reporting the results – please clarify (top of p. 15). These results also appear to imply that the best career advice for women in STEM is to collaborate with individuals from as elite institutions as possible—in that case gender disadvantages appear to be mitigated.

Related to my point above, I recommend that the authors edit the abstract and discussion section of the paper further with the aim of articulating their findings more clearly. Given the complexity of the analyses, I understand that is not easy to do, but distilling what exactly they did (and did not) find would strengthen the paper and also would enable readers who are less familiar with the technical aspects of the analyses to understand the implications of the results. Relatedly, I recommend editing the title to make it more descriptive and to reflect what the authors discovered (as opposed to what they investigated).

I am puzzled by the authors' response to my suggestion to examine the gendered make-up of the co-author pairs. The authors report that they decided not to conduct this analysis due to negative responses on social media about a previous paper reporting discouraging results for female-female protégé/mentor pairs. I still recommend examining either the effect of gender of the senior author and/or the gender make-up of the pairs on the outcomes of interest.

Response to the reviews of manuscript NCOMMS-21-50002A-Z: “Untangling the network effects of productivity and prominence among scientists”

Dear Editors and Reviewers,

Thank you again for our detailed attention to our manuscript, from technical concerns to broad ideas. Your comments have led us to revise text and several figures of the paper for clarify and detail, include a richer body of literature, and insert additional statistical analyses and robustness tests. While the results of the paper have not qualitatively changed, the paper is now stronger. We have also carefully followed the editorial checklist about paper formatting, reporting summaries, and prepared other required forms. We hope that you find this final revision improved as a result of the changes, and will consider recommending it for publication.

Our responses are organized into the following sections:

- Response to Reviewer 1
- Response to Reviewer 2
- Response to Reviewer 3
- Response to Reviewer 4

Sincerely,

Weihua Li, Sam Zhang, Zhiming Zheng, Skyler J. Cranmer, and Aaron Clauset

Response to Reviewer 1

I recognize the enormous effort made by the authors to defend their position and to get around the real essence of productivity definition, without pondering on the implications. "Not everything that counts can be counted, and not everything that can be counted counts." The authors have chosen to count what does not count. It is true, unfortunately, that many bibliometricians perpetuate the cozy definition of productivity as the number of publications by an individual, which makes its measurement easy "to count". The question is: "does it count?". Would you promote or allocate more research funds to those who publish more or to those who have higher impact on the advancement of science, all inputs being equal? It is not simply a matter of definitions, it is the meaningfulness of definitions that matter. The authors could simply change the wording substituting productivity with intensity of publication, to be conceptually correct, but the essence would be the same.

What should we do then with the findings of the effects of networking on the intensity of publication alone? I could go on with the invalid full counting of publications, and their gross field classification, which add to the original sin. Probably others can accept the authors' approach, but I cannot: science has to move forward, it cannot stay still (and above all wrong).

Addressed: We thank the Reviewer again for the thoughtful comments and concerns in the previous round, and we believe that addressing them has substantively strengthened the connection to past literature and improved discussion of policy and practices. Norms in academic performance evaluation have been shifting over the past decades as a result of evolving practices in many aspects of science such as research methods and team formation. As the sheer volume of publications and the sizes of teams consistently increase over time, allocating credit among team members based on contribution and impact has become more important and controversial than ever. Untangling the network effects of the two most commonly used and generally accepted metrics, i.e., productivity and prominence, allows us to effectively understand the roles of social networks in shaping academic careers and intensively unpack socialization induced inequalities in science. Researchers in the field of scientometrics and the science of science have been proposing new measures of productivity by accounting for exogenous factors like impact, and our approach would further enhance their efforts in understanding the roles of social networks in science.

Moreover, to clarify our usage of the metrics, we now define productivity and prominence in the Abstract, and define high-impact when we are introducing prominence in the Introduction. We also discuss the limitations of the selected metrics in the Introduction and Discussions, so that researchers and policy-makers would be cautious in using them. We hope

that this would help the reader understand our research motivations and rule out ambiguity and misunderstandings.

Response to Reviewer 2

The research is timely and provides a novel contribution to the literature. The findings that network effects have more influence on publishing quantity and impact than other variables like gender and institutional prestige support emerging work in this area. The perspective of the literature review, and interpretation of the results, are consistent with recent research showing collaboration networks can largely explain differences in publishing rates. The current manuscript advances this argument by also looking at article impact (citation rates), beyond the mere quantity of publication. There is also more refined data analysis in excluding low-rate publishers and focusing on mid-career scientists and their senior collaborators. Further, the current paper controls for confounding variables.

I believe the authors have adequately addressed concerns I raised in the first review.

Addressed: Thank you again for your insightful comments in the previous review report that has substantively helped to improve our manuscript!

Response to Reviewer 3

The authors have exhaustively addressed my initial comments, which were largely of clarifying nature.

To reiterate my initial assessment, this work is an exhaustive analysis on careers in science that integrates various analysis, modeling and visualization methods in exemplary form, and as such, merits publication.

Addressed: We thank the Reviewer again for providing insightful comments in the earlier review report that helped us to substantively improve the literature review, statistical analysis, and visualizations of our manuscript!

Response to Reviewer 4

Overall, the authors have responded thoroughly to my comments about the previous version of the manuscript. I appreciate their comprehensive responses and additional analyses/results reported.

Addressed: We thank the Reviewer again for the thoughtful comments and recommendations of a number of robustness checks in the previous round, which we believe that addressing them has substantively improved the quality of our manuscript. Next, we offer a detailed point-by-point response to your additional suggestions.

The results concerning the interaction effects of gender by institutional prestige were quite interesting. I didn't follow what the authors meant by the last sentence in the new paragraph reporting the results – please clarify (top of p. 15). These results also appear to imply that the best career advice for women in STEM is to collaborate with individuals from as elite institutions as possible—in that case gender disadvantages appear to be mitigated.

Addressed: We have edited the last sentence of the new paragraph as the Reviewer suggested. Our finding that prestige does not appear to drive gendered inequalities in researchers' productivity is in line with other recent studies, as they find that this gap is largely explained by prestigious departments offering more funded research labor. However, although these results do appear to imply that women in STEM could benefit in collaborations with researchers from elite institutions, we would still be cautious to offer any policy recommendations based on the finding, because it might have side-effects in encouraging collaborations with individuals from non-elite institutions.

Related to my point above, I recommend that the authors edit the abstract and discussion section of the paper further with the aim of articulating their findings more clearly. Given the complexity of the analyses, I understand that is not easy to do, but distilling what exactly they did (and did not) find would strengthen the paper and also would enable readers who are less familiar with the technical aspects of the analyses to understand the implications of the results. Relatedly, I recommend editing the title to make it more descriptive and to reflect what the authors discovered (as opposed to what they investigated).

Addressed: The Reviewer makes a good suggestion here. To better clarify our usage of metrics, we have defined productivity and prominence in the abstract and introduction. We

have also included a less technical summary paragraph to concisely explain the implications of our findings in discussions: *“We find that the observed gendered gap in productivity and prominence can be largely explained by differences in social networks. The way social networks can behave like social capital, with boosting effects on junior researchers decaying as their senior collaborators age. After controlling for network effects, our adjusted productivity and prominence parameters can explain a significant proportion but not all scholarly disparity related to environmental prestige. These results have implications for gendered and institutional differences in scholarship, which we discuss further in the following paragraphs.”*

While we agree with the Reviewer that further editing the languages to make the narrative more descriptive would facilitate readers less familiar with the methodology to understand our findings, we believe our work would also attract audience from other scientific communities like network science, so retaining some features of networks narrative would help them understand our work too.

I am puzzled by the authors’ response to my suggestion to examine the gendered make-up of the co-author pairs. The authors report that they decided not to conduct this analysis due to negative responses on social media about a previous paper reporting discouraging results for female-female protégé/mentor pairs. I still recommend examining either the effect of gender of the senior author and/or the gender make-up of the pairs on the outcomes of interest.

Addressed: We thank the Reviewer for proposing this analysis and agree that the results would be important for the scientific community. However, we still believe that due to the complexity and sensitivity of this analysis of gendered pairing among coauthors, it would merit a separate, in-depth study that goes beyond the scope of this paper.